# Prior information differentially affects discrimination decisions and subjective confidence reports

Marika Constant [1,2,3] ✉, Michael Pereira [4], Nathan Faivre [4] &
Elisa Filevich [1,2,3,5]

According to Bayesian models, both decisions and confidence are based on the same precision-weighted integration of prior expectations ("priors") and incoming information ("likelihoods"). This assumes that priors are integrated optimally and equally in decisions and confidence, which has not been tested. In three experiments, we quantify how priors inform decisions and confidence. With a dual-decision task we create pairs of conditions that are matched in posterior information, but differ on whether the prior or likelihood is more informative. We find that priors are underweighted in discrimination decisions, but are less underweighted in confidence about those decisions, and this is not due to differences in processing time. The same patterns remain with exogenous probabilistic cues as priors. With a Bayesian model we quantify the weighting parameters for the prior at both levels, and find converging evidence that priors are more optimally used in explicit confidence, even when underused in decisions.

Human perception has often been shown to be based on a Bayesian inference process, in which the brain infers information about the environment by integrating incoming information with previous beliefs[1]. Computationally, this involves the integration of a prior distribution ("prior") with a likelihood distribution ("likelihood") to give a posterior distribution ("posterior"), which then forms the basis of a belief or percept. Many studies have shown evidence supporting the idea that perception and decision-making can be explained as Bayesian inference[1–7]. Further, confidence, i.e. the sense of certainty that typically accompanies perceptual decisions, can also be explained by Bayesian inference models.

In formal terms, Bayesian models propose that confidence corresponds to the perceived posterior probability of being correct about our inferences, based on the relative strengths of the posterior probabilities of each hypothesis being considered. This Bayesian confidence model has been highly influential, and it has been tested and supported empirically in both animals and humans[8–14], though

alternatives have been proposed[15–17]. Importantly, this framework considers that the posterior percept is based on the precision-weighted integration of priors and likelihoods, and also that this same posterior gives rise to decisions and confidence. This simple formulation relies on two assumptions: first, that both decisions and confidence integrate priors and likelihoods optimally; and further, that confidence integrates priors and likelihoods in the same way as decisions. But these assumptions must be empirically tested, as the alternative is also possible: either confidence, decisions, or both, might integrate priors in systematically biased ways, and might do so asymmetrically.

Meanwhile, there is evidence to suggest that these kinds of sub-optimalities often occur in human perception and confidence. Various systematic biases have been found in the information that enters confidence, such as a bias towards decision-congruent evidence[18], or an overweighting of perceived sensory noise[19,20]. Furthermore, several studies have found that confidence incorporates different or

[1]Humboldt-Universität zu Berlin, Faculty of Life Sciences, Department of Psychology, Unter den Linden 6, 10099 Berlin, Germany. [2]Bernstein Center for Computational Neuroscience Berlin, Philippstraße 13 Haus 6, 10115 Berlin, Germany. [3]Berlin School of Mind and Brain, Humboldt-Universität zu Berlin, Luisenstraße 56, 10115 Berlin, Germany. [4]Université Grenoble Alpes, Université Savoie Mont Blanc, CNRS, LPNC, 38000 Grenoble, France. [5]Hector Institute for Education Sciences & Psychology, University of Tübingen, Europastraße 6, 72072 Tübingen, Germany. ✉e-mail: marika.constant@gmail.com

additional information compared to decisions[21–26]. This supports the possibility that there are asymmetries in the way that certain sources of information influence these different processing levels. With regard to prior information, empirical work examining Bayesian confidence models has typically used uninformative, 'flat' priors, so it has not been possible to detect these potential biases or asymmetries. Two recent studies have begun examining confidence under informative priors, and found that confidence thresholds liberalise following prior-congruent stimuli[27,28]. These results show that priors influence confidence, but still cannot answer how optimal this influence is, and how it compares to the influence on decisions. In order to understand how our sense of confidence arises across different situations in which we may have highly informative prior expectations, as well as to rigorously test Bayesian confidence models, it is critical to understand quantitatively how priors are weighted relative to likelihoods in confidence computations, and how this relates to their weighting in decisions. More broadly, due to the pervasive role of priors in our processing, studies assessing the Bayesian confidence model under informative priors are important for generalizability.

Here, we examine (1) whether the use of prior information is optimal relative to the use of new sensory information, and (2) regardless of optimality, whether prior information is used the same way at the level of decisions and confidence. We do this both behaviourally and by fitting a generative Bayesian model with free weighting parameters that allows us to quantify the relative use of priors at the level of decisions and confidence. Additionally, we assess whether any possible asymmetries in the use of the prior at these different processing levels could be explained by differences in evidence accumulation time, or by the nature of the task and prior used.

In two (of three) experiments, participants complete a dual-decision task in which they make right/left decisions about two consecutive dot motion stimuli per trial. Critically, participants are informed about the added rule that, following correct responses to the first ('lead') stimulus, the second ('target') stimulus will go to the right. Conversely, incorrect decisions about the lead stimulus will be followed by leftwards-moving target stimuli. This means that, in an optimal Bayesian observer, the prior expectation for a rightward target stimulus will be equal to the decision confidence about the lead stimulus. In other words, if participants are very confident that their response to the lead stimulus is correct, they will have a very strong prior that the target will be a rightward stimulus. Conversely, if participants are very unsure about their response to the lead, they will have equal expectations that the target will go right or leftwards. A dual-decision task with this same rule was used recently[29] to investigate the influence of the prior on the decision level, which was interpreted as a measure of implicit confidence. Here we build on that work in order to assess the potentially differential role of priors in decisions and in explicit, subjective confidence ratings.

In our design, we also make use of the task structure in order to be able to vary the strength of priors and likelihoods on the same dimension. Only this way can we directly, behaviourally compare the relative influence of priors and likelihoods on responses. We do this by building two conditions that are matched in the amount of total available posterior information, but differ in whether the lead or target is more informative. This allows us to measure whether accuracy, confidence, and metacognitive efficiency differ between these two conditions, which can indicate either over- or underweighting of the priors relative to likelihoods. Going further, in order to quantify that weighting in both decisions and confidence, and test the precise way in which the use of prior information might differ at these different processing levels, we fit a Bayesian model to the data with parameters capturing the weighting of the prior in decisions and confidence. In the second experiment, we then further investigate whether potential asymmetries between the weighting of priors in decisions versus confidence could be simply attributed to differences in processing time. Finally, in the third experiment, we test whether the results generalise to a single-decision task with exogenously cued probabilistic priors, which is a common task structure for manipulating prior expectations[27,28].

## Results

### Experiment 1: dual-decision task and conditions

On each trial of the dual-decision task, participants ($N = 21$) saw two consecutive random dot motion stimuli and made a decision after each about whether the coherent motion was to the right or left (Fig. 1a), followed finally by a confidence rating about the second decision. We told participants that their task was to herd a flock of sheep (represented as dots in the RDK stimulus) towards the barn on the right of the screen. We gamified the rule (linking correct lead-decision responses to rightwards-moving target stimuli) and asked participants to position a sheepdog by responding to the lead stimulus. If the sheepdog was in the correct place, the sheep (coherently moving dots) of the target stimulus would go to the right. This rule meant that participants' internal decision confidence about the lead stimulus formed the strength of their prior for a rightward target stimulus. In this way, the strength of the prior could be manipulated by changing the coherence of the lead stimulus (with L: low, M: medium, or H: high coherence), and the strength of the likelihood could be manipulated by changing the coherence of the target stimulus (also L, M, or H). This allowed us to create two conditions with the same available posterior information to the optimal observer ('posterior level'): One in which there was more prior information due to a stronger lead stimulus (Stronger-Lead) and one in which there was more new sensory information due to a stronger target stimulus (Stronger-Target) (Fig. 1b, c). These two conditions existed in matched pairs across three overall levels of available posterior information ($L_{post}$: L + M or M + L, $M_{post}$: L + H or H + L, and $H_{post}$: M + H or M + H). With this experimental design, we were able to assess whether participants' accuracy, confidence and metacognitive performance depended on the condition.

### Manipulation check

We first ensured that response accuracy increased with increasing coherence of the stimuli, indicating that we effectively manipulated internal signal strength as intended. Additionally, we investigated whether participants used the task structure as we wanted them to, using the rule and hence their prior to guide their target decisions, at least to some extent. If this were true, we expected participants to perform better on the target decisions of each trial, on which they had additional information from the prior (lead stimulus) to guide their choice. To test both these predictions we built a logistic regression model on response accuracy with fixed effects of information level (L, M, H), decision order (lead or target), and their interactions (see Table 1 for the model syntax). We found a significant interaction between decision order and information level, $\chi^2(2) = 43.53$, $p < 0.001$, $BF_{10} = 9.71 \times 10^4$, with differences between information levels reduced in the target decisions. However, post-hoc pairwise comparisons revealed that response accuracy remained significantly higher with increasing information levels for both decisions (all $p < 0.001$), $OR_{Lead:Low/Medium} = 0.65$, 95% CI [0.58, 0.74], $OR_{Lead:Medium/High} = 0.64$, 95% CI [0.56, 0.74], $OR_{Target:Low/Medium} = 0.83$, 95% CI [0.72, 0.95], $OR_{Target:Medium/High} = 0.78$, 95% CI [0.68, 0.90], suggesting our coherence manipulation to work as planned (Fig. 2a). Additionally, response accuracy was higher in the target decision compared to the lead decision at each information level, and this was significant at the low and medium information levels (both $p < 0.001$), $OR_{Low:Lead/Target} = 0.61$, 95% CI [0.53, 0.69], $OR_{Medium:Lead/Target} = 0.77$, 95% CI [0.67, 0.88], $OR_{High:Lead/Target} = 0.94$, 95% CI [0.81, 1.09], revealing appropriate use of the task structure.

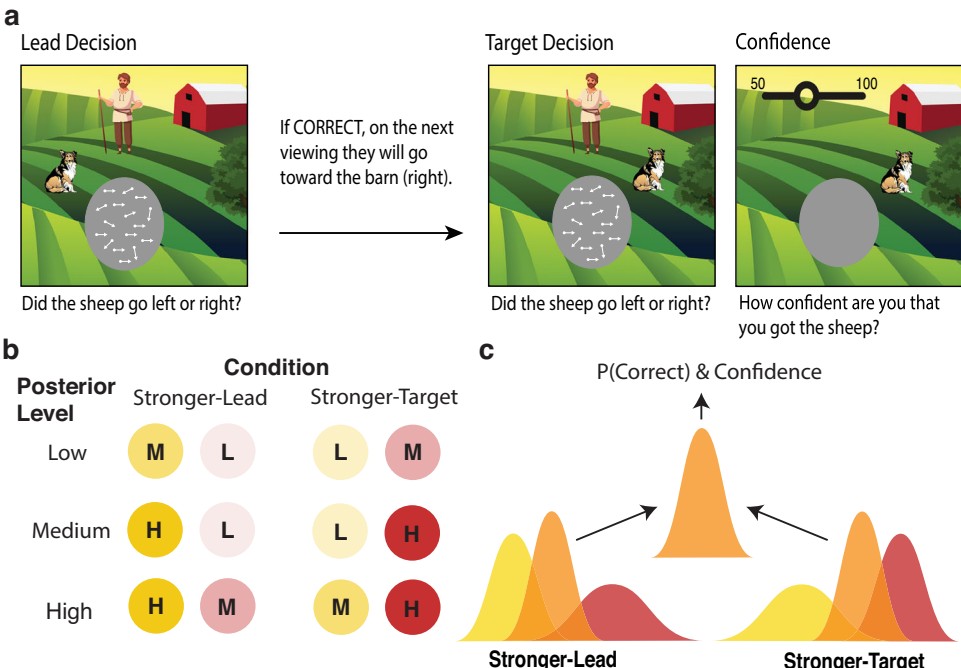

**Fig. 1 | Task and conditions sketch. a** Gamified dual-decision paradigm. On each trial, participants viewed and made right/left decisions about two consecutive dot motion stimuli (lead and target), which they were told represented flocks of sheep. We explicitly informed participants that if they were correct about the first decision, then the target stimulus would be going to the right, and if incorrect, then it would be going to the left. This meant that, in an optimal observer, the prior for a rightward target stimulus should be equal to the lead decision certainty. They also rated their confidence in the target decision. In Experiment 2, the paradigm was the same except there was a 2-s delay after viewing the target stimulus, before participants were allowed to make the target decision. **b** Conditions. We manipulated the coherence of the lead and target stimuli (each of which could have L: low, M: medium, or H: high coherence), here depicted with the circle transparency, to create two conditions that had matched available posterior information but differed in whether the lead or target stimulus was stronger. This was tested at three overall posterior levels - low posterior information (L + M), medium posterior information (L + H), and high posterior information (M + H). **c** Sketch of the Stronger-Lead vs Stronger-Target manipulation. The posterior percept (orange) should optimally be the precision-weighted integration of the prior (yellow), which in our task was always rightward and could span from 50–100%, and likelihood (red). Both conditions led to the same available amount of posterior information, which (in the optimal case) leads to the same probability of a correct choice as well as confidence. Hence, the target accuracy and confidence will only differ between conditions if the two sources of information are not integrated optimally in the decisions and/or confidence. Note that this is only a sketch aimed at conveying the intuition of how the conditions were matched in terms of posterior information. The prior for a rightward target stimulus is captured more accurately by a step function, shown in Fig. 8a.

## Dependence on condition reveals suboptimal prior weighting in decisions

After ensuring that the manipulations worked as intended, we investigated the use of prior information in decisions. We reasoned that, if the prior were suboptimally weighted at the decision level, performance on the target decision would depend on the condition despite matched posterior information (Fig. 1c). To evaluate this, we built a logistic mixed effects model on target decision response accuracy. This included fixed effects of posterior level (L, M, H), condition (Stronger-Lead, Stronger-Target), and their interaction (Table 1). We did not find a significant interaction effect. In line with our manipulation check, we found a significant main effect of posterior level, $\chi^2(2) = 175.77$, $p < 0.001$, $BF_{10} = 6.12 \times 10^{36}$, with accuracy increasing with higher available posterior information (Fig. 2b), $OR_{Low/Medium} = 0.71$, 95% CI [0.64, 0.80], $OR_{Low/High} = 0.53$, 95% CI [0.47, 0.59], $OR_{Medium/High} = 0.74$, 95% CI [0.65, 0.83]. We also found a significant main effect of condition, $\chi^2(1) = 9.06$, $p = 0.003$, $BF_{10} = 8.95$, with better performance when the target was stronger, $OR_{Stronger-Lead/Stronger-Target} = 0.89$, 95% CI [0.82, 0.96], $Z = -3.09$, $p = 0.002$ (Fig. 2b). This suggests that, despite an equal amount of available posterior information, participants performed better when more of that information was carried by the target stimulus, rather than the lead stimulus, indicating that they dismissed some of the prior information when making their decisions. In other words, this revealed an underweighted use of the prior at the decision level, relative to an optimal observer. Because there was potential for response bias to lead to differences in accuracy across the conditions regardless of how the prior information was weighted, we reran this model, including a measure of response bias obtained from fitting subject-wise psychometric functions to performance on the unbiased first decisions (see Methods). This did not substantially change the result or conclusions, as there were no significant effects involving response bias.

## Analysis of mean confidence reveals less underweighting of priors than in decisions

Before looking at the use of prior information in confidence, we investigated whether confidence followed the commonly found 'folded-X' pattern when collapsed across the Stronger-Lead and Stronger-Target conditions. This would involve an interaction on confidence between response accuracy and evidence strength. We built a linear mixed effects model on confidence with fixed effects of posterior level (L, M, H), target response accuracy (Correct, Incorrect), and their interaction. This revealed the expected interaction between posterior level and response accuracy, $F(2,14842) = 24.5$, $p < 0.001$, $BF_{10} = 2.84 \times 10^6$, $M_{Diff(Correct-Incorrect): Low} = 6.52$, 95% CI [5.18, 7.86], $M_{Diff(Correct-Incorrect): Medium} = 9.37$, 95% CI [7.98, 10.76], $M_{Diff(Correct-Incorrect): High} = 10.66$, 95% CI [9.23, 12.09], replicating the folded-X confidence pattern. We then built a linear mixed effects model on confidence with fixed effects of posterior level (L, M, H), condition (Stronger-Lead, Stronger-Target), target response accuracy (Correct, Incorrect), and their interactions (Table 1). We found

## Table 1 | Regression Models

| Analysis | Hypothesis | Model Formula |
|---|---|---|
| Manipulation Check | Accuracy depends on information level and on which decision it is (Lead Decision vs Target Decision). | logit(Response Accuracy) ~ Information Level * Decision Order + (1|Participant) |
| Type 1 Analysis | Target accuracy depends on posterior information level and, if priors are used suboptimally, on the condition (Stronger-Lead vs Stronger-Target). | logit(Target Response Accuracy) ~ Posterior Level * Condition * Response Bias + (1|Participant) |
| Type 2 Analysis | If priors are used suboptimally, condition modulates the effect of posterior information on confidence differently following correct and incorrect target decisions. | Confidence ~ Posterior Level * Condition * Target Response Accuracy * Response Bias + (Target Response Accuracy + Condition|Participant) |

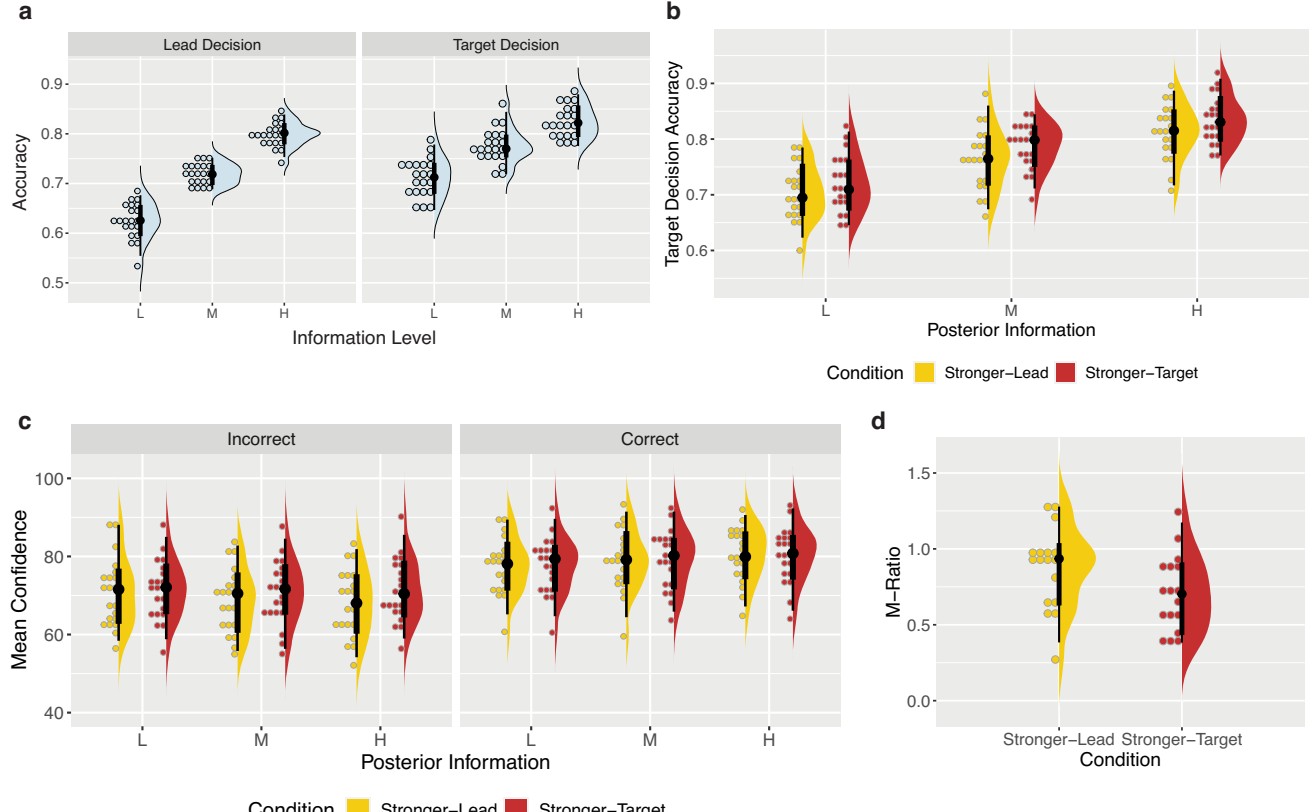

**Fig. 2 | Behavioural results.** In the raincloud plots of all panels, the right-half, split violin plots correspond to the probability density, and the vertical black lines correspond to the median, IQR. The hinges show the first and third quartiles, and the vertical whiskers show +/- 1.5IQR. The binned dotplots on the left half show each individual subject ($N = 21$) as a point. **a** Manipulation check. As expected by experimental design, increasing the information level increased accuracy in both the lead decisions ($M_{L\_Lead} = 0.63$, SD = 0.04; $M_{M\_Lead} = 0.71$, SD = 0.02; $M_{H\_Lead} = 0.80$, SD = 0.02) and target decisions ($M_{L\_Target} = 0.71$, SD = 0.04; $M_{M\_Target} = 0.78$, SD = 0.03; $M_{H\_Target} = 0.82$, SD = 0.03), suggesting that changes in dot coherence successfully increased the strength of the internal signal. Accuracy was also higher in the target decisions compared to the lead decisions, suggesting at least some use of the task structure and prior information. **b** Effect of condition on accuracy. We found a significant effect of the condition (which stimulus was stronger) on accuracy, despite matched available posterior information, suggesting participants to give suboptimal relative weights to priors and likelihoods in their decisions. Participants performed significantly better in the Stronger-Target condition ($M_{L\_Stronger-Target} = 0.72$, SD = 0.05; $M_{M\_Stronger-Target} = 0.79$, SD =

0.04; $M_{H\_Stronger-Target} = 0.84$, SD = 0.04) compared to the Stronger-Lead condition ($M_{L\_Stronger-Lead} = 0.70$, SD = 0.05; $M_{M\_Stronger-Lead} = 0.76$, SD = 0.05; $M_{H\_Stronger-Lead} = 0.81$, SD = 0.05), pointing to a relative underweighting of prior information. **c** Effect of condition on confidence. We found no effect of condition on confidence following correct trials (with differences between conditions—Stronger-Target minus Stronger-Lead—of: $M_L = 0.15$, SD = 2.19; $M_M = 0.40$, SD = 2.34; $M_H = -0.07$, SD = 2.15), but an effect of condition following incorrect trials, with lower confidence in the Stronger-Lead condition (with differences between conditions—Stronger-Target minus Stronger-Lead—of: $M_L = 0.66$, SD = 2.51; $M_M = 1.98$, SD = 2.50; $M_H = 3.15$, SD = 2.54). From the decision level as a baseline, this result indicates greater use of the prior in confidence than in decisions. **d** M-Ratio estimates. M-Ratios per participant for each condition, measured across all posterior levels. As expected if (as **b** and **c** suggest) valid prior information informed confidence ratings more than discrimination decisions, M-Ratios were significantly higher in the Stronger-Lead condition than in the Stronger-Target condition. Source data are provided as a Source Data file.

evidence for a three-way interaction from Bayesian statistics $BF_{10} = 70.83$ (Fig. 2c) but not from frequentist tests ($F(2,14832) = 2.66$, $p = 0.070$). Post-hoc pairwise comparisons revealed a significant effect of condition on confidence only for incorrect trials at the high posterior level, $t(1281) = -3.52$, $p < 0.001$,

$BF_{10} = 6.14$, $M_{Diff(Stronger-Target\ –\ Stronger-Lead)} = 3.15$, 95% CI [1.45, 4.85], and medium posterior level, $t(839) = -2.16$, $p = 0.030$, $M_{Diff(Stronger-Target\ –\ Stronger-Lead)} = 1.74$, 95% CI [0.23, 3.24], although the Bayesian statistics suggest evidence against the effect at the medium level ($BF_{10} = 0.46$). There were no significant effects of the

condition following correct trials, with evidence for the null hypothesis at all posterior levels (all BF < 0.02). Because the sub-optimal weighting of the prior at the decision level can impact confidence distributions, a statistically significant effect of condition on confidence cannot be interpreted in isolation. Instead, we must consider the decision level result as a baseline. Simulations shown below revealed that, given any underweighting of the prior in decisions, the following pattern holds: If the prior was even further underweighted in confidence than in decisions, we would expect confidence in the Stronger-Target condition to be higher following correct trials and lower following incorrect trials. Contrary to this, the result of no difference following correct trials and lower confidence in the Stronger-Lead condition following incorrect trials indicates that the prior is less underweighted than at the decision level, and is hence used more optimally in confidence. To quantify this effect and concretely compare the weighting of the prior at the decision versus confidence level, we used computational modelling. Further in line with this idea, we found significantly higher M-Ratios in the Stronger-Lead condition ($M = 0.87$, SD = 0.27) than in the Stronger-Target condition ($M = 0.71$, SD = 0.25), t(16) = 3.64, $p = 0.002$, $\eta^2_p = 0.01$, 95% CI [0.14, 1.00] (Fig. 2d), suggesting that the relative use of priors and likelihoods is more optimal at the meta-cognitive level than at the level of decisions.

## Bayesian model fits different weights of priors in confidence and decisions

Overall, the behavioural analyses revealed that, while decisions underweighted the available prior information, confidence seemed to use the prior information more optimally. To account for these results, and to be able to compare the effects at these two levels of processing more directly and quantitatively, we use a computational model. Additionally, the models we fit to the data do not rely on the matched pairs of posterior information, and can hence provide important converging lines of evidence to validate the model-free results in case the conditions were not perfectly matched in their internal processing. We built and fit a Bayesian model of decisions and confidence under informative priors that included a weighting parameter for the prior (relative to the likelihood), for both the decision ($w_{choice}$) and the confidence rating ($w_{conf}$). These weighting parameters reflected potential over- or underestimation of the precision of the prior, hence weighting its influence over behaviour relative to the likelihood. They were also computationally identical in their impact, scaling the perceived variance of the prior, allowing us to directly compare them. We first estimated internal noise and decision bias (not under the influence of a prior) per participant on separate data using the Akaike-weighted combination of four fit psychometric functions (see Methods), and then incorporated these subject-wise estimates into the model, allowing us to account for them. The model additionally included a parameter for confidence bias ($b$) that captured overall over- or underconfidence in both stimuli. The model had three free parameters (per participant as well as at the group level for a hierarchical implementation): $w_{choice}$, $w_{conf}$, and $b$. We checked that the parameters were recoverable in a parameter recovery analysis (Supplementary Information, Fig. S4). We also compared our full model, which we call the Flexible model, to a variety of simpler models, to assess each main research question in more depth.

At the decision level, the model predicted that if the prior was optimally weighted relative to the likelihood, ($w_{choice} = 1$), there would be no difference in accuracy between the two conditions (Fig. 3a, left panel). If the prior was overweighted ($w_{choice} < 1$, capturing an underestimation of the variance of the prior), accuracy would be higher when the lead was the stronger stimulus (Fig. 3a, middle panel). Conversely, if the prior was underweighted ($w_{choice} > 1$), accuracy would be higher when the target was the stronger stimulus (Fig. 3a, right panel). We first fit the full model individually to each participant, limiting

complexity and allowing us to avoid the simplification needed for a hierarchical implementation (see below). We found a mean $w_{choice} = 2.42$ (SD = 2.09) across participants. This suggests, in line with the behavioural results, that priors are underweighted in decisions, relative to an optimal observer. The simulated choice behaviour for these model results is shown in Fig. 3b (right panel) and demonstrates higher accuracy when the target is the stronger stimulus, just as we found behaviourally. To analyse this formally at the group level, making use of the Bayesian model fitting approach, the full hierarchical model was fit. This model fit the data with a mean of the posterior for the group mean parameter $w_{choice} = 2.17$ (SE = 0.007; Fig. 3c), again suggesting the underweighting of the prior in decisions.

We then examined the model predictions for confidence, given the result of the underweighted priors in decisions ($w_{choice} > 1$). With overweighting of the prior in confidence ($w_{conf} < 1$), the Stronger-Lead condition tends to show higher confidence than the Stronger-Target condition for correct trials, and lower for incorrect trials (Fig. 4a, middle-left panel). In other words, $w_{conf} < 1$ will lead to differences in mean confidence between correct and incorrect that are larger in the Stronger-Lead condition than in the Stronger-Target condition. In contrast, with underweighting of the prior ($w_{conf} > 1$), it is the Stronger-Target condition that tends to have larger differences in mean confidence between correct and incorrect (Fig. 4a, middle-right panel). If the prior weighting in both decisions and confidence were optimal, the model predicts no difference in mean confidence between conditions.

Importantly however, $w_{conf}$ cannot be interpreted in isolation, and must be always considered together with $w_{choice}$. That is, it is useful to focus on the difference $w_{choice}-w_{conf}$. In particular, the pattern of results that we described above for the case of $w_{choice} = 1$ and $w_{conf} < 1$ is similar to $w_{choice} > 1$, and an optimal $w_{conf} = 1$, as both cases correspond to $w_{choice} > w_{conf}$. The model predicts more extreme confidence (higher for correct, lower for incorrect) in the Stronger-Lead condition (Fig. 4a, far-left panel). The behavioural pattern seen in confidence—of no difference between conditions following correct trials, and lower confidence following incorrect trials in the Stronger-Lead condition—lies somewhere in between the patterns predicted by optimal weighting in confidence (where $w_{conf} \ll w_{choice}$) and by equal underweighting in confidence as in decisions ($w_{conf} = w_{choice}$), and is hence captured when the prior is less underweighted in confidence than in the decision ($w_{conf} < w_{choice}$) (Fig. 4a, far-right panel).

Fitting the full model individually to each participant yielded a mean $w_{conf} = 1.97$ (SD = 1.81) across participants. Out of these 20 participants, 15 had a fit $w_{conf}$ value that was smaller than their fit $w_{choice}$ value, indicating stronger weighting of the prior in confidence than in decisions. We did an initial comparison of these individually fit $w_{choice}$ and $w_{conf}$ parameters per participant using a paired samples t-test, which demonstrated that the $w_{choice}$ parameters were significantly larger than the $w_{conf}$ parameters ($M_{Diff}=0.46$, $SD_{Diff} = 0.69$), t(19) = 2.97, p = 0.008, $BF_{10} = 6.25$, $\eta^2_p = 0.32$, 95% CI [0.06, 1.00]. This, in line with the behavioural results, suggests the weighting of the prior in confidence to be closer to optimal, compared to the weighting of the prior in decisions. The simulated confidence for these model results (Fig. 4b) successfully captures the pattern associated with the combination of an underweighted prior in decisions and a less underweighted prior in confidence (Fig. 4a, far-right panel).

To assess the weighting of confidence formally at the group level using the hierarchical model, due to complexity constraints, we required a simplification. We modelled confidence on each trial assuming that the internal signal corresponded to the mean external stimulus strengths from each stimulus coherence level (L, M, H). That is, we did not fit the internal latent samples that are theoretically generated from the external stimuli, as that would have led to an overparameterized model with two free parameters per individual trial (totalling close to 30000 free parameters). However, this meant that for the simplified model, stimuli would always lead to internal samples

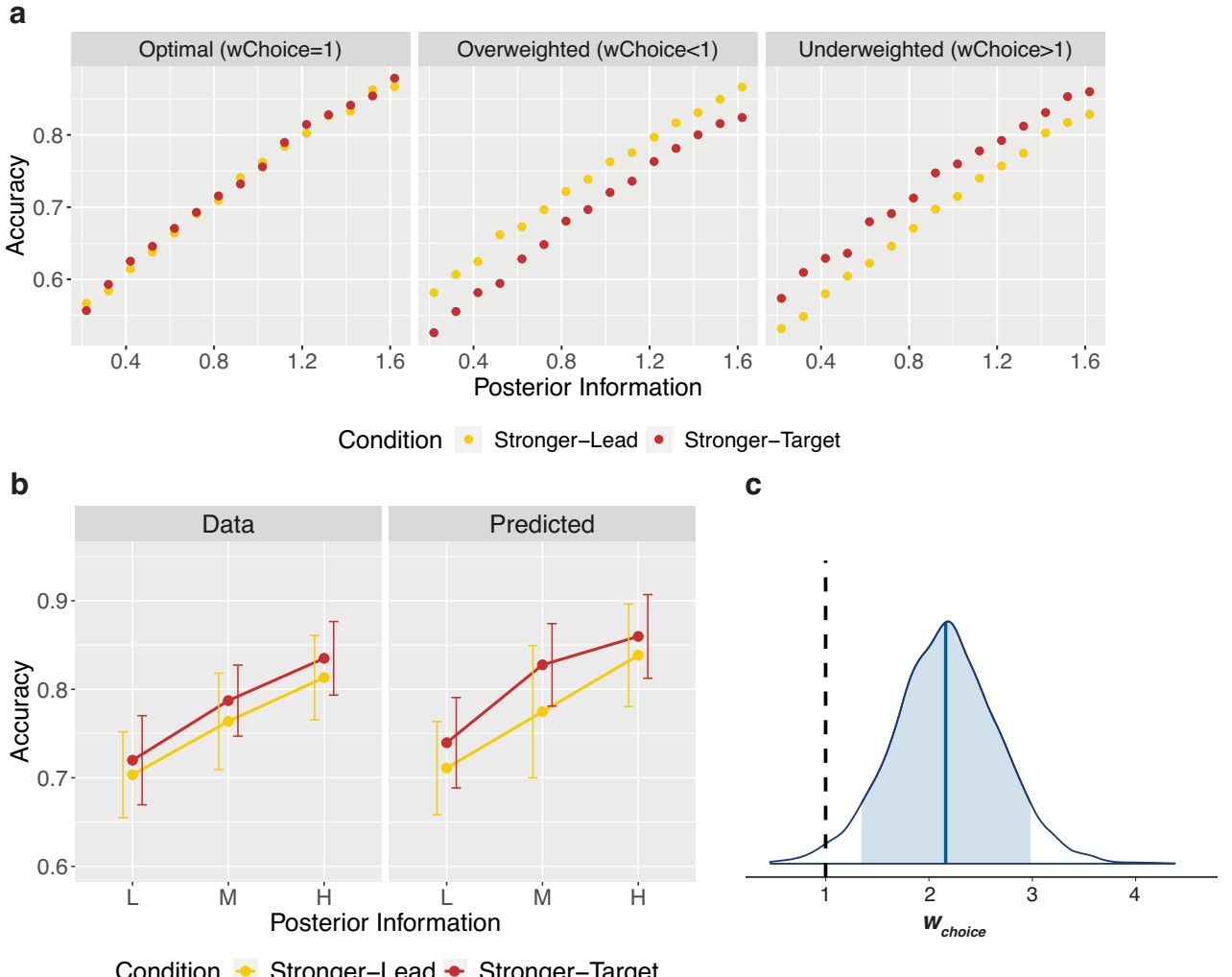

**Fig. 3 | Modelling discrimination decisions. a** Model simulations of decision accuracy. Target decisions were simulated from the Flexible model across different posterior levels, between the two conditions, and at three different values of $w_{choice}$: 1 (optimal weighting of prior information), 0.33 (overweighting of prior), and 3 (underweighting of prior), shown from left to right respectively. These values are representative of the range found in the data, and capture either over- or under-estimated variance by a factor of 3. The resulting decision accuracies are shown here. The model predicts that optimally using prior information will lead to no difference in accuracy between the two conditions, whereas overweighting prior information will lead to higher accuracy when the lead is stronger, and under-weighting prior information will lead to higher accuracy when the target is stronger. **b** Data and predictions of the individually fit model. The left panel shows the

mean observed accuracies per posterior level and condition. The right panel shows the predicted accuracies generated from sampling each individual participant's fit posterior parameter distributions 10 times and simulating 720 trials for each of those sampled parameters, also using that participant's staircased coherences, internal noise and decision bias. Error bars capture standard deviation (SD) of accuracies across participants ($N$ = 20). **c** Hierarchical model posterior distribution for $w_{choice}$. The posterior distribution for the group mean parameter of the weighting of prior information in the decision, $w_{choice}$. The blue shaded region shows the 89% credible interval and the vertical black dashed line reflects optimal weighting of the prior in the decision ($w_{choice}$ = 1). A weight above 1 captures overestimation of the variance and hence underweighting of the prior. Source data are provided as a Source Data file.

on the correct side of the decision criterion, leading the model to underestimate confidence overall. This predicted underestimation of confidence is described in detail in Supplementary Information, but critically, the simplification still allowed the model to capture differences between conditions, and hence our parameter of interest, $w_{conf}$. Additionally, we showed that any impact that this could have had on $w_{conf}$ worked directly against our conclusions (see Supplementary Information), making our interpretations more conservative. The fit hierarchical model had a mean posterior of the group parameter $w_{conf}$ = 1.27 (SE = 0.002), and of the group bias parameter $b$ = 2.18 (SE = 0.003; Fig. 4c), further supporting the results from the non-simplified model.

Finally, with this hierarchical modelling approach, with both the Flexible model and simpler models, we were able to more directly compare the relative weighting of priors and likelihoods at the

decision and confidence level (Fig. 5a–c). We found that $w_{choice}$ was credibly different from $w_{conf}$, with the difference distribution just excluding 0 in the 89% credible interval [0.03, 1.77] (Fig. 5b). This again suggests the weighting of priors relative to likelihoods to differ at the level of decisions and confidence.

**Flexible model describes behaviour better than simpler models**
We then compared the Flexible model to simpler models (Fig. 5a). We checked that these models were distinguishable from one another with a model recovery analysis (see Supplementary Information, Fig. S6).

In addition to the (1) Flexible model, we built (2) the Flat Prior model in which the lead stimulus information was not used at all and the target stimuli were considered to occur under a flat prior ($w_{choice}$ and $w_{conf}$ infinitely large), (3) the Optimal model in which $w_{choice}$ and

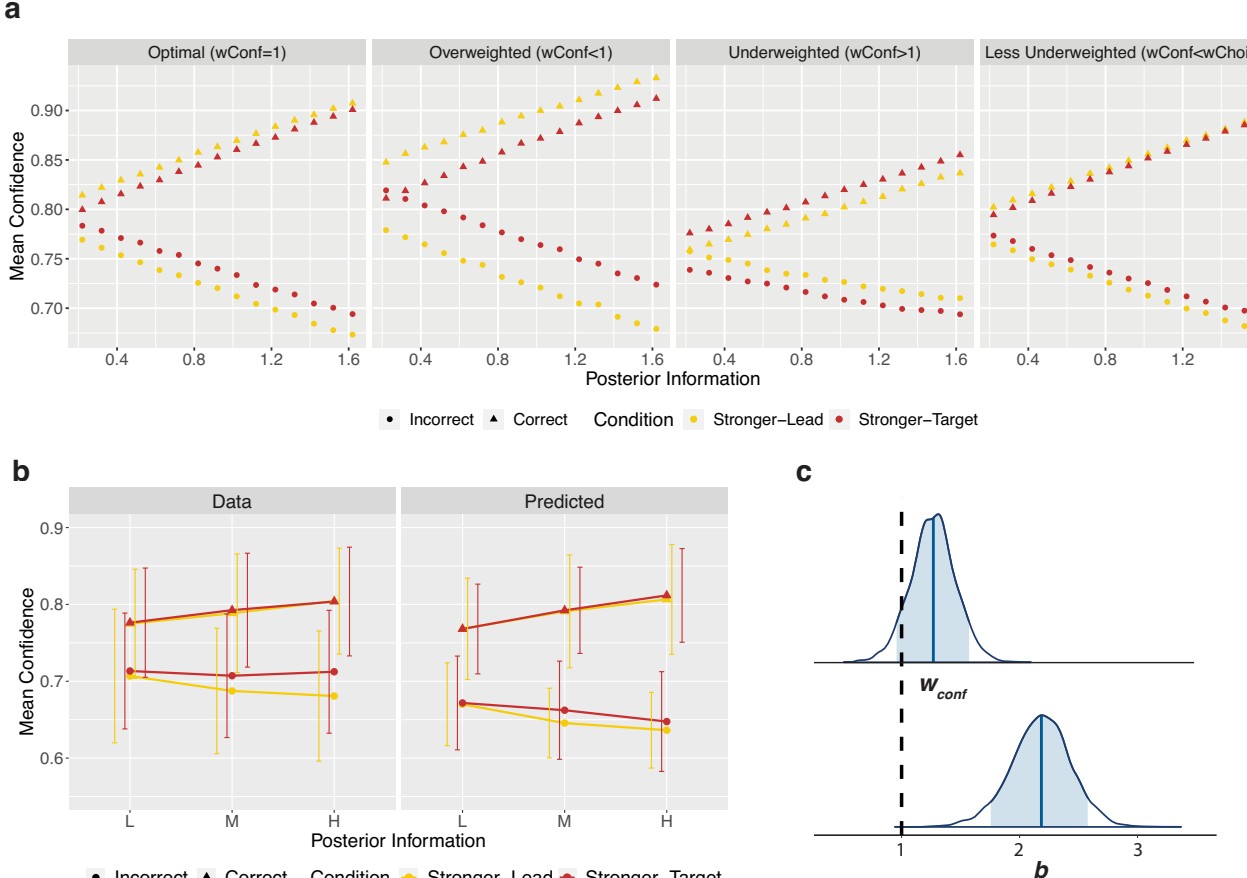

**Fig. 4 | Modelling Confidence. a** Model simulations of confidence. Confidence ratings following correct and incorrect trials were simulated from the model across different posterior information levels, between the two conditions. Simulations used the $w_{choice}$ value from the model fit, 2.17, and four different values of $w_{conf}$: 1 (optimal weighting of prior), 0.33 (3-fold overweighting of the prior), 3 (3-fold underweighting of the prior), and 1.27 (the value obtained from the model fit, underweighting the prior in confidence less than in the decision), shown from left to right respectively. The resulting mean confidence values are shown here. The model predicts mean confidence to increase with increased available posterior information following correct decisions and decrease with increased available posterior information following incorrect trials. As the prior is increasingly overweighted, the model predicts higher mean confidence in the Stronger-Lead condition following correct decisions, and lower in the Stronger-Lead condition following incorrect decisions (middle-left panel). As the prior is increasingly underweighted, the opposite pattern is predicted (middle-right panel). Due to the suboptimal weighting at the decision level, the model predicts differences in mean

confidence when $w_{conf}$ is optimal (far-left panel). When the prior is underweighted in confidence, but less so than in the decision, the model can produce the pattern seen behaviourally (far-right panel). **b** Data and predictions of the individually fit model. On the left are the data, showing confidence following correct and incorrect decisions per posterior level and condition. On the right is mean confidence generated from sampling each individual participant's fit posterior parameter distributions 10 times and simulating 720 complete trials for each of those sampled parameters, also using that participant's staircased coherences, internal noise and decision bias. Error bars capture standard deviation (SD) of mean confidence across participants ($N = 20$). **c** Hierarchical model posterior distribution for $w_{conf}$ and $b$. The top posterior distribution is for the group mean parameter of $w_{conf}$. The lower posterior distribution is for the group mean parameter of $b$. The blue shaded regions show the 89% credible intervals and the vertical black dashed line corresponds to the parameter values of an optimal observer. Source data are provided as a Source Data file.

$w_{conf}$ were both optimal (equal to 1), and (4) the Equal model in which the prior was forced to be used to the same extent in decisions and confidence ($w_{choice} = w_{conf}$) and hence only one $w$ parameter was fit. All models included confidence bias as a free parameter. We fit these models and compared their predictive performance against the Flexible model using a 10-fold leave-one-group-out cross-validation (LOGO-CV, where 'groups' correspond to participants). The Flexible model predicted the data better than each of the other models, with a difference in expected log pointwise predictive density (elpd_diff) of −619688.2 (se_diff=16717.0) compared to the Flat Prior model, of −11017.7 (se_diff=4534.4) compared to the Optimal model, and of −76685.8 (se_diff=2147.4) compared to the Equal model (Fig. 5c). The results of this formal model comparison are in line with both the behavioural and the modelling results shown above: The difference to the Flat Prior model suggested that participants used the prior information to some extent, in line with our behavioural manipulation

check. The difference to the Optimal model suggested that participants used the prior information suboptimally. And, the difference to the Equal model suggested that participants used the prior information to a different extent in confidence than in decisions, in line with our finding of a credible difference between $w_{choice}$ and $w_{conf}$.

**Experiment 2: dual-decision task with delayed target decision**
The finding from Experiment 1, that priors can be used more optimally in confidence, might support the idea that priors are integrated gradually, and that there is continued post-decisional evidence accumulation that can then factor into confidence. This would be in line with previous work suggesting that confidence computations incorporate additional information that has accumulated after the first-order decision[21–24,26,30,31]. Alternatively, it is possible that the additional use of the prior in confidence is by virtue of the introspective act, and not simply due to continued evidence accumulation. In support for the

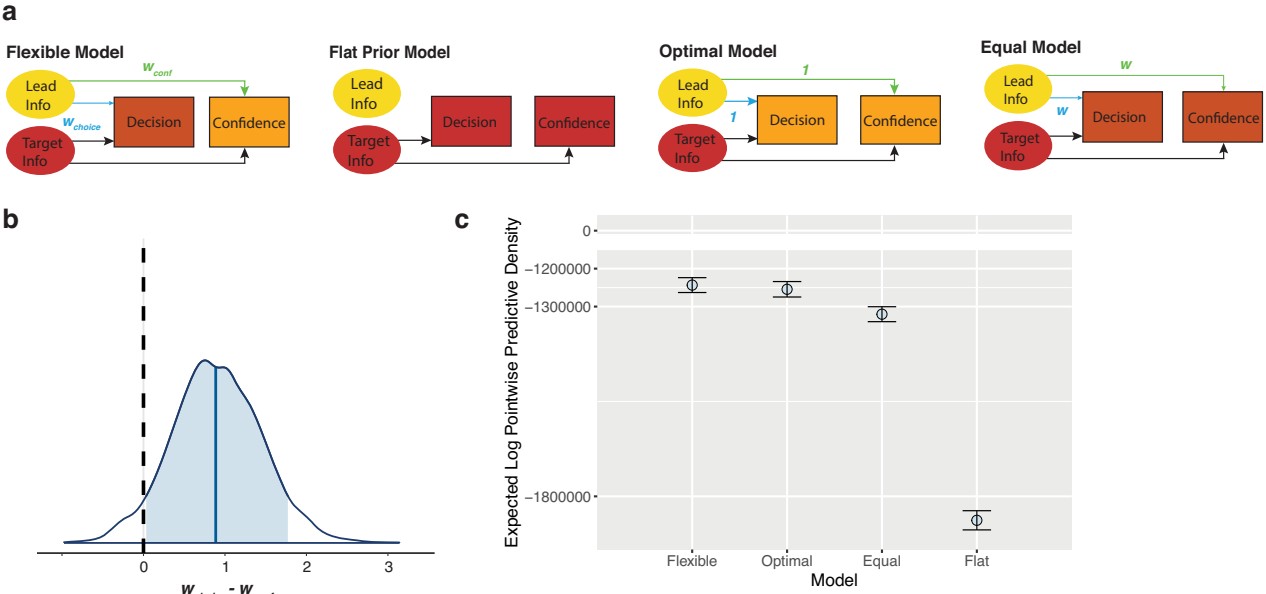

**Fig. 5 | Model comparison results. a** Schematic of the four compared models. Schematics of the four models we compared, with the thickness of the light blue arrows depicting the weighting of priors relative to likelihoods in the decision and the thickness of the light green arrows depicting the weighting of the priors relative to likelihoods in confidence. In the Flexible Model, relative weighting of priors and likelihoods was allowed to differ between decisions and confidence ($w_{choice}$ was allowed to differ from $w_{conf}$). The Flat Prior Model only has arrows from the target, showing that the prior information was not used at all in decisions or confidence. In the Optimal Model, $w_{choice}$ and $w_{conf}$ both had to be optimal (equal to 1). In the Equal Model, the weighting of priors relative to likelihoods was allowed to differ from optimal but the pattern was the same between $w_{choice}$ and $w_{conf}$. **b** Posterior group difference distribution of $w_{choice} - w_{conf}$. The posterior distribution for the

difference in the group mean parameters $w_{choice}$ and $w_{conf}$. The blue shaded region shows the 89% credible interval and the vertical black dashed line reflects no difference in the two parameters ($w_{choice} - w_{conf} = 0$). 0 is just excluded from the 89% credible interval, suggesting $w_{choice}$ and $w_{conf}$ to be credibly different from one another. **c** Expected Log Pointwise Predictive Density (ELPD) results from LOGO-CV. The predictive capacity of each model is shown as the elpd value from the LOGO-CV with 10 folds, leaving out and then predicting 2 participants per fold. This shows the Flexible Model to have the highest predictive capacity, suggesting it as the best model to explain the data. This is followed by the Optimal Model, Equal Model and then Flat Prior Model. Error bars depict SE. Source data are provided as a Source Data file.

latter alternative, previous work found enhanced metacognitive efficiency following prior congruency, showing information from priors to especially boost metacognitive judgements[28]. In order to investigate these two possibilities further, we ran a second pre-registered experiment ($N = 25$) in which we repeated the same paradigm under the same conditions, but we added a 2-s delay after the offset of the target stimulus and before participants were allowed to report their target decision. We chose the duration of the delay to approximately match the peak of the distribution of reaction times between viewing the target stimulus and giving the confidence rating from Experiment 1, which was 2.48 seconds. Therefore, if in Experiment 1 the more optimal use of the priors in confidence was only due to the extra processing time before giving the confidence rating, then delaying the target decision until that time point in Experiment 2 should lead to more optimal use of the prior information in the delayed target decision. If, however, the more optimal use of the prior was due to the introspective confidence rating, delaying the target decision in Experiment 2 should not change the pattern of results.

**Priors underweighted in decisions despite increased processing time**
We ran the same regression model on target response accuracy as in Experiment 1 and found a significant interaction between posterior level and condition, $\chi^2(2) = 8.21$, $p = 0.017$, $BF_{10} = 2.68$. In line with Experiment 1, response accuracy was higher in the Stronger-Target condition at each posterior level, although this was only significant at the medium posterior level, $z = -4.30$, $p < 0.001$, $BF_{10} = 138.49$, $OR_{Medium:Stronger-Lead/Stronger-Target} = 0.76$, 95% CI [0.63, 0.92] (Fig. 6a). This suggests the prior to be underweighted in the target decisions, even after the delay and hence the added opportunity for evidence

accumulation (for behavioural data, see Supplementary Information, Fig. S1). At the confidence level, we found a significant main effect of condition, $F(1,17923) = 16.84$, $p < 0.001$, $BF_{10} = 34.92$, with confidence increasing in the Stronger-Target condition following both correct and incorrect trials, $M_{Diff(Stronger-Target\ -\ Stronger-Lead):Correct} = 0.57$, 95% CI [0.19, 0.94], $M_{Diff(Stronger-Target\ -\ Stronger-Lead):Incorrect} = 1.13$, 95% CI [0.41, 1.85]. This is slightly different from Exp. 1, where we found an interaction between the condition and response accuracy. However, due to the finding of strongly underweighted prior in decisions, the behavioural predictions at the confidence level and hence the interpretation of these results become less clear. To examine it precisely, and test whether the same pattern of more optimal use of priors in confidence remained, we again fit the Flexible model, this time as confirmatory modelling analyses. Due to further complexity constraints, here we only used the hierarchical model with the simplification. Replicating the pattern found in Experiment 1, we found a mean of the posterior of $w_{choice} = 3.97$ (SE = 0.02; Fig. 6a) and a mean of the posterior of $w_{conf} = 1.39$ (SE = 0.01; Fig. 6b), with a credible difference between them [0.50, 4.64] (Fig. 6c). Taken together, this suggests that prior information is underweighted in decisions even when those decisions occur at a similar time point as the confidence judgements in Experiment 1, and, like in Experiment 1, confidence then more optimally uses that prior information.

**Experiment 3: single-decision task with cued probabilistic priors**
In Experiments 1 and 2, we used the dual-decision task in order to allow us to manipulate the strength of the prior and the likelihood on the same scale. This made it, at least in principle, possible to create the matched conditions, allowing us to do quantitative behavioural analyses along with the modelling. However, this comes at the cost of

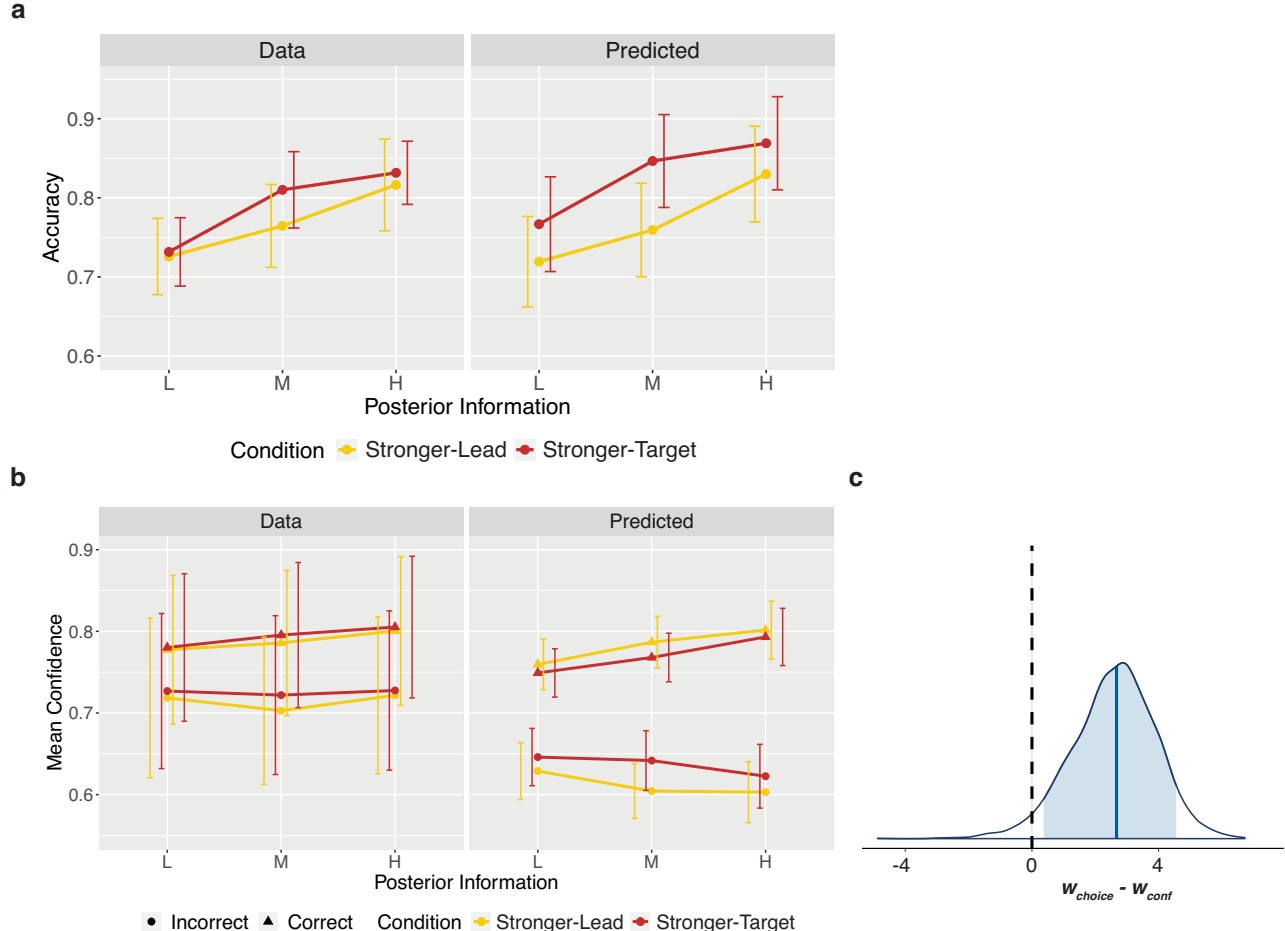

**Fig. 6 | Experiment 2 results. a** Decision level model results against data. The left panel shows the observed accuracies per posterior level and condition (*N* = 25 participants). The right panel shows the predicted accuracies generated from sampling the fit posterior group mean parameter distributions 1000 times, and simulating 720 trials per participant for each of those sampled parameters, also using each participant's staircased coherences, internal noise and decision bias. Error bars capture standard deviation (SD) of accuracies across participants. **b** Confidence level model results against data. On the left are the data, showing confidence following correct and incorrect decisions per posterior level and condition (*N* = 25 participants). On the right is mean confidence generated from sampling the fit posterior parameter distributions 1000 times and simulating 720 complete trials per participant based on those parameters, also using each participant's staircased coherences, internal noise and decision bias. Note that here, unlike in Fig. 4b, we show results from the hierarchical model for which, due to the necessary simplification, the underestimated confidence is expected. However, this does not interfere with our interpretation of the differences between conditions or $w_{conf}$. Error bars capture standard deviation (SD) of mean confidence across participants. **c** Posterior group difference distribution of $w_{choice} - w_{conf}$. The posterior distribution for the difference in the group mean parameters $w_{choice}$ and $w_{conf}$. The blue shaded region shows the 89% credible interval and the vertical black dashed line reflects no difference in the two parameters ($w_{choice} - w_{conf} = 0$). 0 is excluded from the 89% credible interval, suggesting $w_{choice}$ and $w_{conf}$ to be credibly different from one another. Source data are provided as a Source Data file.

potentially reducing the generalizability to a more standard exogenous cue task. Hence, we ran a third experiment (*N* = 20) using exogenous probability cues as priors. In this task, participants were explicitly told the probability of either a rightward or leftward stimulus, after which they viewed the dot motion stimulus, made their single decision, and then rated their confidence. The prior probabilities were always true, and corresponded to either a 0.6, 0.7, 0.8, or 0.9 probability of the coherent motion going towards either the right or left, counterbalanced across participants. There were also some trials without an informative prior (0.5 probability of left versus right) which were used to measure decision bias, internal noise, and metacognitive noise, as these were necessary for the modelling (see Supplementary Information).

We first investigated the behavioural results qualitatively by comparing them to predictions from model simulations. If the priors are used in decisions, we would expect a positive relationship between the strength of the prior for a rightward stimulus and the probability of choosing right. As the priors are more underweighted, this relationship

gets weaker (Fig. 7a). Looking at the results, we found a weaker relationship than would be expected of optimal prior use (Fig. 7b), suggesting underweighting of priors in decisions. If priors are used in confidence, we would expect higher confidence with stronger rightward and leftward priors, leading to a U-shaped relationship between confidence and rightward priors from 0.1–0.9. As the priors are more underweighted in confidence, this relationship flattens. Because the expected patterns at the confidence level depend on the decision level results, we simulated prior weighting in confidence that was either equal to, stronger than, or weaker than the decision level (Fig. 7c). Comparing the data to these predictions revealed a stronger U-shaped relationship between confidence and rightward prior strengths than would be expected of equal weighting (Fig. 7d), suggesting that confidence used the priors more strongly than in decisions.

To formally assess this, we adapted and then fit the hierarchical Flexible model to the data, including the same free weighting parameters and confidence bias parameter (see Supplementary Information for details of the model adaptations to suit this task, and note that

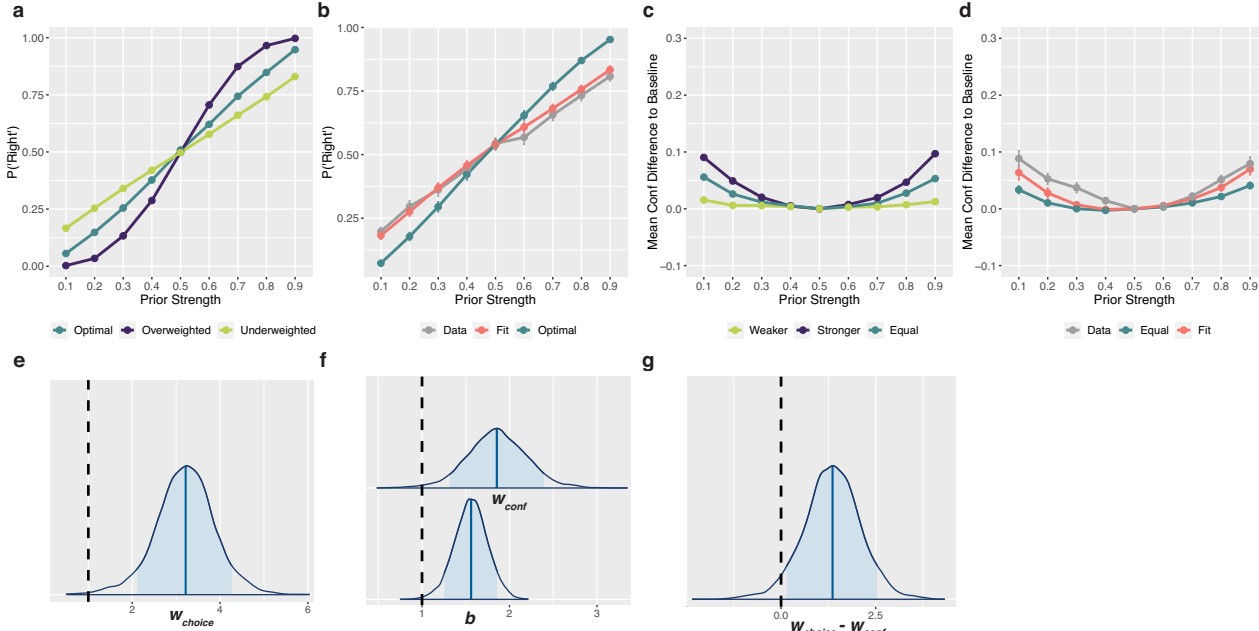

**Fig. 7 | Experiment 3 results. a** Decision level predictions. The probability of choosing right (P("R")) given each rightward prior, simulated with three levels of prior weighting: optimal ($w_{choice} = 1$), overweighting ($w_{choice} = 0.5$), and under-weighting ($w_{choice} = 2$). The model predicts that weaker prior weighting in the decision leads to a weaker relationship between the rightward prior strength and the P("R"). **b** Decision level results. Simulations from the fit model shown against the data, with optimal weighting as a reference. This reveals a weaker slope of the P("R") across rightward prior strengths compared to optimal, indicating under-weighting of priors. **c** Confidence level predictions. Confidence given each level of rightward prior, simulated with three levels of prior weighting relative to decision level underweighting ($w_{choice} = 2$): equal ($w_{conf} = 2$), stronger ($w_{conf} = 1$), and weaker ($w_{conf} = 4$) weighting. The model predicts that prior weighting in confidence influences the curvature of the relationship between confidence and prior strength. To highlight that curvature, confidence is shown normalized to the flat prior condition (Prior Strength = 0.5) as a baseline by taking the difference in mean confidence to that baseline. **d** Confidence level results. Simulations from the fit model shown

against the data, with symmetrical weighting ($w_{conf} = w_{choice}$) shown as a reference. This reveals stronger concavity across rightward prior strengths compared to equal weighting, indicating stronger prior weighting in confidence than in decisions. **e** Posterior distribution for $w_{choice}$. The posterior distribution for the group mean parameter of prior weighting in decisions, $w_{choice}$. For **e–g**, blue shaded regions show 89% credible intervals. The vertical dashed line reflects optimal prior weighting ($w_{choice} = 1$). **f** Posterior distribution for $w_{conf}$ and $b$. The top posterior distribution is for the group mean parameter of $w_{conf}$. The lower posterior distribution is for the group mean parameter of $b$. The vertical dashed line corresponds to optimal observer parameter values. **g** Posterior group difference distribution of $w_{choice}$-$w_{conf}$. The posterior distribution for the difference in the group mean parameters $w_{choice}$ and $w_{conf}$. The vertical dashed line reflects no difference in the two parameters ($w_{choice}$-$w_{conf}$ = 0). 0 is just excluded from the 89% credible interval, suggesting that $w_{choice}$ and $w_{conf}$ are credibly different from one another. Source data are provided as a Source Data file.

the model simplification was not required here). Replicating the patterns found using the dual-decision paradigm, we found a mean of the posterior of $w_{choice} = 3.20$ (SE = 0.01; Fig. 7e) and a mean of the posterior of $w_{conf} = 1.86$ (SE = 0.008; Fig. 7f), with a credible difference between them [0.15, 2.54] (Fig. 7g). We found a mean of the posterior of $b = 1.56$ (SE = 0.003; Fig. 7f). This suggests that, even outside of the dual-decision setup and using an exogenously cued prior, prior information is underweighted in decisions and then used to a greater extent in confidence about those decisions.

## Discussion

In three experiments, we tested whether prior information influences confidence optimally, and how this compares to its influence on perceptual decisions. To do so, in the first two experiments, we compared pairs of conditions that were matched in the available posterior information but differed on whether the stronger source of information was the prior or the new incoming information. We then evaluated the differences between conditions (Stronger-Lead vs Stronger-Target) in both discrimination accuracy and mean confidence, and fit a quantitative model to measure the weighting of prior information. This revealed that priors are underweighted relative to likelihoods in discrimination decisions. Conversely, confidence judgements incorporated prior information to a greater extent than discrimination decisions did. Further, and in line with the idea that prior information is processed more optimally at the metacognitive level compared to the

level of first-order decisions, we found that metacognitive efficiency was higher when more information was carried by the prior, with first-order performance hindered while confidence preserved the use of this information. Taken together, these results suggest that we can access and use information from priors in explicit, introspective confidence judgements even to a greater extent than we use that information to guide decisions. This pattern also generalised to a third experiment that moved away from the dual-decision setup and used a task structure commonly found in the literature.

These findings go against the assumptions, implicit in the Bayesian framework, of optimal and equal integration of priors in decisions and confidence. While participants may not necessarily be expected to behave as Bayesian optimal observers, these findings quantify precisely in which way they deviate from those assumptions. Although the underweighted prior in decisions may, in isolation, be explainable by a decay of the prior information over time, such a decay would make the asymmetry between the decision and confidence levels even more surprising, as the confidence judgements occurred even later. The results of Experiment 2 revealed, further, that this asymmetry remains even when additional processing time is given by forcing a delay before the target decision. This suggests that this pattern cannot be accounted for just by continued evidence accumulation before the confidence rating, since a similar amount of evidence accumulation should have occurred between the confidence ratings in Experiment 1 and the target decisions in Experiment 2. Instead, this points towards a

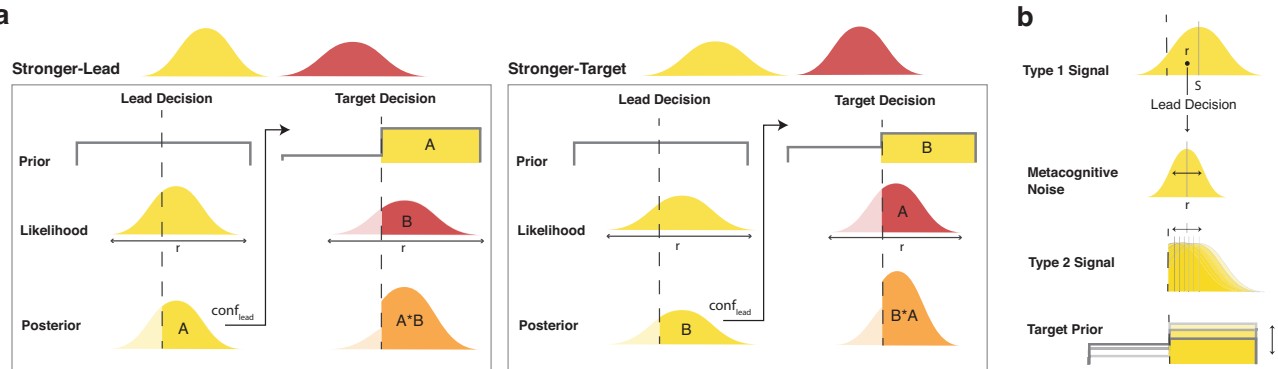

**Fig. 8 | Model schematic. a** Model of conditions. In the model, lead decisions occurred under a flat prior. The lead stimulus generated an internal response with added internal noise, forming the likelihood and then posterior distribution. The area (A in Stronger-Lead condition, B in Stronger-Target condition) of the posterior on the chosen side of the decision boundary (vertical dashed line) equals confidence in the lead decision ($conf_{lead}$). $conf_{lead}$ formed the strength of the prior for a rightward target stimulus. The target stimulus generated an internal response, forming the target likelihood, which integrated with the prior to give the target posterior (orange). This posterior led to the target decision and confidence rating. The strength of the posterior probability of the winning hypothesis (opaque orange area) is based on the combination of the prior strength (either A or B, depending on condition) and likelihood strength (either B or A, depending on condition). Because these combine to the same posterior strength (A*B or B*A), the model predicted equal accuracy and mean confidence between conditions, given optimal relative weighting of priors and likelihoods. The weighting parameters, $w_{choice}$ and $w_{conf}$, scaled the estimated variance of the lead, effectively scaling $conf_{lead}$, and captured the strength of prior that best explains target decisions and confidence ratings, respectively. **b** Metacognitive noise in the model. In the dual-decision paradigm, the target prior is susceptible to metacognitive noise, since it is based on an internal confidence computation. Metacognitive noise adds noise to the first-order internal response $r$ (from the Type 1 distribution) for the confidence computation. Since $r$ is the mean of the Type 2 distribution on which confidence is computed, this jitters the resulting confidence value (or the yellow area under the distribution—area A in the leftmost panel of this figure). Note that this does not bias confidence, but makes it more variable across trials, and less reflective of accuracy. Since the rightward target decision prior equates to the confidence value on each trial, metacognitive noise also makes the strength of the rightward target decision prior more variable across trials, not biased overall.

more optimal use of prior information at the metacognitive level, compared to first-order processing. In line with this, Balsdon and colleagues demonstrated asymmetries in the information used by decisions and confidence[25,26]. They used series of stimuli and found decisions to set covert bounds and stop collecting new evidence, while confidence used more of the available information. In light of those findings, our work shows a similar effect, where decisions make use of less of the information available than confidence does. However, here, we find that it is prior information that is more strongly dismissed in decisions.

These results add an additional layer to recent findings by Lisi et al.[29], who focussed on implicit, not explicit confidence. They, as we did, found priors to be underweighted at the level of the decision, but could not assess whether explicit confidence weighted them differently than decisions. Here we reveal important differences in how the prior is used at different processing levels, by examining the weighting of the prior in explicit confidence as well. Our results suggest that, even though the prior is underweighted in a decision, people can access and use this information better when asked to make an explicit introspective judgement about that decision. A cognitive architecture in which perceptual decisions can primarily respond to current incoming evidence while higher order metacognitive processing integrates different sources of information and monitors their relative certainty might be highly adaptive. For example, it might be beneficial to react rapidly and in accordance with evidence for even an unlikely belief if that would pose some threat, meanwhile having the metacognitive system accurately track its posterior probability for appropriate models of the world. Our results therefore provide crucial insight on a dissociation between human behaviour and associated confidence.

We now consider our findings in light of several potential alternative explanations. First, given the nature of the dual-decision setup, participants might behave in a confirmatory way: they might be biased towards target decisions that support their lead decision. Although this was not in line with the rule of the dual-decision task, other work has found people to be confirmatory even at a perceptual level and it is still possible that participants gathered evidence in this way[32–34]. To assess this, we investigated whether the performance and confidence on the target decision was higher when the target direction matched the lead direction, as would be predicted of this confirmatory behaviour. This revealed participants not to have such a confirmatory bias, and in fact to have a slight effect in the opposite direction, though this repulsion effect also failed to explain the pattern of results (see Supplementary Information). Next we consider alternative explanations due to the nature of the prior as an internal confidence value. A 'confidence leak' effect has been found in the literature, which involves confidence from a previous decision influencing upcoming confidence[35]. It is possible that the internal confidence from the lead decisions therefore 'leaked' in this way to the target confidence. This would not influence the decision level results, but would cause target confidence to be more extreme in the Stronger-Lead condition, since that condition would be associated with higher internal lead confidence. However, these are not the patterns we see in the data, and even when combined with the decision level result, this cannot reproduce the asymmetry found between the decision and confidence levels (see Supplementary Information). Finally, we consider the role of metacognitive noise[36] in our interpretations. Because the prior is an internal confidence value in the dual-decision setup, computing it could involve metacognitive noise (Fig. 8a, b). This noise would make the estimated priors less reflective of the true priors, but would not cause consistent over- or underestimation of the prior (Fig. 8b). Therefore, this effect would not look like over- or underweighting of priors, and would not cause differences between conditions. Still, to further explore how this interacts with the model, we simulated three different implementations of both Gaussian and log-normally distributed metacognitive noise (Supplementary Information). None of these influence the differences between conditions or capture the pattern of results. In sum, these other possible effects of the dual-decision task and prior used in Experiment 1 and 2 cannot capture the findings. Instead, the results can best be explained by a suboptimal and asymmetrical weighting of the prior: underweighting of the prior in decisions and less underweighting of the prior in confidence.

The results of Experiment 3 also, importantly, showed this dissociation to generalise outside of the context of a dual-decision task and of a prior that is based on a confidence computation. Using exogenously cued probabilistic priors, participants still underweighted the prior in decisions and used the prior more strongly in confidence. This is a common task structure in work assessing the role of priors in decisions, so the findings here should be considered carefully when interpreting other results. For example, substantial other work has found that probabilistic priors are underused in decisions, with participants failing to shift their decision criterion as much as would be optimal[37–45]. However, our findings add an additional, critical layer to these results, revealing that participants do still process and have access to that ignored prior information, but that the use of this information occurs at the metacognitive level. Therefore, a more complete picture of the influence of priors on perceptual decision-making should include confidence.

In this study, both paradigms used informative, high level priors. Future work is necessary to investigate whether this result holds true when different kinds of priors are used. First, lower-level priors such as the light-from-above prior, cardinal orientation bias, or perceptual history bias might affect decisions differently, as they may act at an earlier stage and impact perception of the target stimulus more directly[46–48]. Second, non-informative or suboptimal priors might reveal that the pattern we see here reflects a confidence bias towards prior information, rather than more optimal use of priors in confidence. If so, in cases of suboptimal priors, confidence would still be more likely to be affected by the invalid prior information than decisions. This possibility is in line with recent studies that have shown that confidence is biased by suboptimal, false priors about stimulus precision[49] or about task performance[50]. Other work testing the Bayesian confidence model has found confidence to suboptimally overweight evidence that is in line with the decision, leading to a form of confirmation bias in perceptual confidence[18,32,51]. Our findings do not show a confirmation bias in favour of information in line with the decision, but might rather reflect a confidence confirmation bias in favour of information in line with the prior, even in cases where this actually contradicts the decision. Although at face value this may go against the previous confirmation bias findings in perceptual confidence, this might, speculatively, still be in line with the conclusions drawn, namely that confidence favours evidence consistent with one's beliefs. This could also be a strategy aimed at self-consistency and avoiding cognitive dissonance[51–53], leading people to be more confident in response to information that fits in their belief system, and to doubt themselves when they act against their prior world models.

## Methods

The first two experiments were pre-registered (Experiment 1: https://osf.io/qgpsr, 25.10.21 and Experiment 2: https://osf.io/tvyrz, 02.08.22), and we respected the pre-registered plan unless stated otherwise.

### Participants

For Experiment 1, we pre-registered that we would test 25 participants across two sessions. We chose this sample size to be close to previous studies using similar tasks and modelling methods[8,29], which included between 15 and 26 participants. We also pre-registered six minimal criteria to invite participants to the second session. The most important of these criteria were (1) that response accuracy increased across the three coherence levels—hence suggesting that the experimental manipulation had the intended effect on internal signals −, and (2) that response accuracy was (any amount) higher on the second decisions as compared to the first, indicating basic use of the task structure. Following these criteria, we excluded 12 participants without inviting them to take part in the second session, and collected data until we reached 25 participants that met these criteria and were tested for two sessions. Four of these participants were later excluded from analysis

because they no longer met these basic criteria after including data from their second session, leaving a total of 21 participants (10 male, 11 female) included in the analyses. Participants were tested in Berlin, were healthy and were between 18 and 37 years of age (M = 25.7, SD = 4.6). For all experiments, participants' gender was determined based on free-form, optional self-reporting. No sex or gender-based analyses were performed and we did not consider sex or gender in the study design, as neither sex nor gender played a role in our research questions. Participants all reported to have normal or corrected-to-normal vision, were fluent in English, and primarily right handed (Edinburgh Handedness Inventory score: M = 83.2, SD = 28.5). Participants were compensated with 8€ per hour or with equivalent course credit and gave signed, informed consent before starting the experiment. The ethics committee of the Institute of Psychology at the Humboldt-Universität zu Berlin approved the study (Nr. 2021-47), which conformed to the Declaration of Helsinki.

For Experiment 2, we pre-registered that we would test an initial 25 participants that met the minimal exclusion criteria after both sessions, after which we set a stopping rule, based on evidence for or against the effect of condition on target response accuracy. After 25 participants, we found substantial evidence for the alternative hypothesis and stopped collecting data. These 25 participants included 9 male, 15 female, and 1 that did not specify; were between 19 and 34 years of age (M = 25.4, SD = 3.8); and were primarily right handed (Edinburgh Handedness Inventory score: M = 86.3, SD = 34.3—note one participant was excluded from this due to missing data), as well as meeting the same inclusion criteria as in Experiment 1.

For Experiment 3, based on sample sizes for Experiment 1 and 2, we planned to include 20 participants with clean data. We set a priori that participants would be excluded if they performed worse in the 90% prior condition (in which they have the most available information) than in the 50% prior condition (in which they have the least available information), as this would indicate failure to understand the basic task structure. We tested 23 participants (4 male, 19 female) and excluded 3 for that reason. This left a total of 20 participants included in the analysis. They were between 19 and 33 years of age (M = 23.7, SD = 3.9), and were primarily right handed (Edinburgh Handedness Inventory score: M = 85.4, SD = 39.0).

### Setup

The experiment was programmed in HTML/Javascript/CSS to run in the browser. We used JATOS[54] to store the result data. The study ran on Google Chrome (version 94.0.4606.71) on a Dell Precision 5760 laptop (Intel core i7 with 31GB of RAM) with a display resolution of 1,920 × 1,200 (refresh rate = 60 Hz).

### Procedure

**Control task.** In each of the two sessions of Experiment 1 and 2, prior to starting the main task, participants first completed 90 trials of a control task. In Experiment 3, which was only one session, the second round of this control task was done at the end of the session. Each trial of the control task consisted of a single dot motion stimulus with a 50% chance of the coherent motion going to the right vs left, followed by a right/left decision, which participants made using the "S" or "A" keys, respectively. The stimuli in the control task spanned six different coherences, meant to capture a broad range of difficulties - 5%, 10%, 12%, 15%, 20% and 30% coherence. The resulting data were later used to estimate participants' internal noise and decision bias (see below). In total across the two sessions, participants completed 180 control task trials (30 per coherence level). In only Experiment 3, this control task additionally included a confidence rating after each decision, on the same continuous scale as the main task. This was then used to estimate metacognitive noise, which was necessary for the full modelling analyses of Experiment 3. After the control task, we explained the

instructions for the full task structure to participants both verbally and in written instructions, and they then completed five demo trials to familiarise themselves with the task and buttons, and then proceeded to the main task.

**Main task: experiment 1 and 2.** On each trial of the main task, participants completed two consecutive decisions consisting of a random dot motion stimulus followed by a right/left decision using the "S" or "A" keys (Fig. 1a). The first (lead) stimulus of each trial had a 50% chance of the coherent motion going to the right vs left. The direction of coherent motion of the second (target) stimulus depended deterministically on the response accuracy of the first decision such that if they were correct, the second stimulus would have coherent motion to the right, and if they were incorrect, it would have coherent motion to the left. Participants were informed of this rule and instructed to use this to help them in the task. Following an optimal strategy, this conditional rule meant that participants should expect a rightward second stimulus with a strength of prior expectation proportional to their decision confidence about the first decision. Following the second decision, participants rated their confidence on a continuous sliding scale from 50% (guessing) to 100% (totally sure), using the mouse. The dual-decision task structure with the conditional rule was an extension of a previous study investigating implicit confidence[29].

The task was gamified to make it more engaging. The background of the screen was an illustration of fields with a barn to the right, and the stimulus display circle in the middle of the screen (Fig. 1a). The first decision controlled the movement of a cartoon sheepdog to the right or left of the stimulus display, and the second decision controlled the movement of a cartoon farmer to the right or left of the stimulus display. We explained to participants that the moving dot stimuli depicted flocks of sheep, with some "leader" sheep that moved coherently to either the left or right, and that they had to decide based on the motion direction of the leader sheep whether to send their sheepdog to the left or right. Participants were explicitly informed of the rule: If they were correct, and hence the sheepdog was in the correct place, the sheep would be herded toward the barn and on the next stimulus, they would be going to the right. If, however, they were incorrect and the sheepdog was not in the correct place, the sheep would run away and on the next stimulus, they would be going to the left. After the second stimulus, participants then had to make the final decision to either send the farmer to the barn (to the right) to get the sheep, or to send the farmer to herd them from the fields to the left. They then rated their confidence that the farmer had successfully gotten the sheep. We emphasised to participants that they should use the rule to try to help them with the task, and that they should take time to give as sensitive and meaningful confidence ratings as possible.

We manipulated the coherences of the stimuli to create three stimulus levels, and the coherence levels of each of the two stimuli per trial combined to form three overall posterior levels. The low posterior information level consisted of one low and one medium coherence stimulus (L + M or M + L), the medium posterior information level consisted of one low and one high coherence stimulus (L + H or H + L), and the high posterior information level consisted of one medium and one high coherence stimulus (M + H or H + M). These posterior information levels existed across two conditions, a "Stronger-Lead" condition in which the lead stimulus was stronger, and a "Stronger-Target" condition in which the target stimulus was stronger. The stimulus coherence levels were staircased by staircasing lead stimuli, which were not under the influence of an informative prior, with the medium level staircased via a 2-down-1-up procedure targeting 71% accuracy, and the high level staircased via a 3-down-1-up procedure targeting 79% accuracy. The low level was yoked to the medium staircase, but remained 5% lower in coherence, as there was no N-down-1-up procedure that would target an accuracy between 50% and 71%. The three posterior information levels as well as the conditions were

counterbalanced across each block. Participants received feedback about their performance on the target decisions at the end of each block. Each block consisted of 36 trials, and participants completed 10 blocks per session for a total of 360 trials per session and 720 trials in the experiment. Each session took between 1–1.5 h in total.

**Experiment 2.** The paradigm remained the same in Experiment 2 except that there was an added delay period of 2 s before participants could enter the target decision using the "S" or "A" key, after viewing the target stimulus. After these 2 s, a light grey ring appeared around the viewing circle to indicate to participants that they could now report their decision. Participants were instructed to try and avoid pressing a key prematurely during the delay period, although trials with premature presses were not excluded. Participants received feedback at the end of each block about how many premature presses were made, in order to remind them to limit this.

**Stimuli.** The dot motion stimuli were made using an adapted version of an RDK jsPsych plugin[55]. Stimuli were composed of 100 total moving white dots on a circular grey background. Each dot had a radius of 2 pixels and the background circle had a diameter of 425 pixels, with an aperture diameter of 319 pixels (75% of the circle diameter). The noise dots had constant directions that were randomly sampled, and the coherent dots moved in a constant horizontal direction either to the left or right. All dots moved 2 pixels per frame and had a dot life of 17 frames (i.e., each dot followed their trajectory for 17 frames before being redrawn at a random location). Each stimulus was presented for 300 ms. Although some directional information was possible in the random dots of each stimulus, we checked that this did not lead to an overall bias in any participant, so that stimulus directions remained balanced between left and right (for the lead decision, where they were intended to be 50/50), even with the directional information from the noise dots. None of the decisions or confidence ratings was speeded.

**Main task: experiment 3.** On each trial of the main task in Experiment 3, participants made a single decision about the motion direction of a random dot motion stimulus (the same stimuli as in Experiment 1 and 2). Before viewing the stimulus, participants were explicitly informed of the probability that the stimulus would go towards the barn, which was either always to the left or always to the right, counterbalanced across participants. This probability was shown on the screen as either 60, 70, 80, or 90%, and either an "R" for right or an "L" for left, written on the stimulus display circle. This was always the true probability of the stimulus direction stated. Once participants had read and internalised this prior probability, they pressed space to view the stimulus. They then made their decision and rated their confidence in exactly the same way as in Experiment 1 (see above).

There were 100 trials per informative prior probability level (0.6, 0.7, 0.8, 0.9), and these were interleaved and counterbalanced across 10 blocks of 40 trials each. Additionally, there were 100 trials in which there was no informative prior, with a 0.5 probability of right/left. These were spread across 5 blocks of 20 trials each. The first block was a 'flat prior block' and these then occurred after every two 'informative prior blocks'. The flat prior blocks were used to staircase the stimulus difficulty, using a 2-down-1-up procedure targeting 71% accuracy. The stimulus difficulty used for the informative prior blocks was the end point of the staircase from the previous flat prior block. This was done because performance on the informative prior blocks also depended on the prior weighting, which we were measuring, so we could not adaptively adjust stimulus difficulty on the basis of performance on those trials. Together, this totalled 500 trials of the main task, including 400 with informative priors and 100 without. Participants received feedback about their performance after each block. The experiment took 1.5–2 h in total.

## Analysis

We removed any trials with reaction times >8 s on any decision or confidence ratings.

Our main behavioural hypotheses were tested using the 'lme4' package[56] in R[57] for building linear and generalised linear mixed-effects models. For each regression analysis, we used the most complex random effects structure that converged on the full model[58], which meant deviating from pre-registered random effects for analysing confidence. Model syntaxes can be seen in Table 1. For linear mixed-effects models, we ensured that the assumption of normally distributed model residuals was met through visual inspection of residual plots. All hypotheses were tested using two-tailed tests and an alpha level of 0.05, and reported $\chi^2$ values are based on a comparison of the model of interest and null model with the same random-effects structure. We computed effect sizes for single level models as $\eta^2_p$. We additionally computed Bayes factors for our main hypotheses using the 'BayesTestR' package[59] and using Bayesian models with uniform priors with the 'brms' package[60]. For these Bayesian regressions, we ran 4 chains of 10000 iterations, including 2000 burn-in samples, which gave a total of 32,000 effective samples, and we ensured a R-hat close to 1. To analyse whether the prior was suboptimally weighted at the decision level, we deviated from our pre-registered regression approach of examining the effect of condition on the probability of choosing "right" given rightward stimuli. We realised from later simulations that this would not sufficiently distinguish between an optimal and suboptimal weighting of the prior. We instead examined the effect of condition on response accuracy, which could adequately address this question.

In order to compare participants' metacognitive efficiency between conditions, we used the M-Ratio measure (meta-d´/d´) described in previous work[61], with R scripts available from https://github.com/craddm/metaSDT. Two participants were removed from this analysis due to extreme confidence distributions, with over 40% of trials at 100% confidence. Two further participants were removed due to Type 1 hit rates above 0.95 in either condition, but the results did not change when these two participants were included. For measuring M-Ratio, we transformed participants' continuous confidence ratings to a 5-bin discrete scale using quantiles, computed on all ratings per participant.

## Modelling

To quantitatively assess how participants weighted the prior in their decisions and confidence, we fit a Bayesian model to their data. The specific model definitions, model fitting, model selection and evaluation of model results were all exploratory and the details were not pre-registered. The full model, which we refer to as the Flexible model, includes two free weighting parameters, $w_{choice}$ and $w_{conf}$, that capture the weighting of the prior relative to the likelihood in the decision and in confidence, respectively. These parameters act in the same way in the model, scaling the estimate of the variance of the prior, and can hence be directly compared. The model also takes as input a measure of the internal noise and decision bias per subject, which were fit independently using psychometric functions. The hierarchical model implementations were fit to all participants' trial-wise decisions and confidence ratings together and all models were fit using a Markov chain Monte Carlo (MCMC) approach in STAN[62] and with the 'cmdstanr' package[63]. R-hat values were close to 1 (<1.1) for all parameters. One outlier participant was removed from the hierarchical modelling due to a $w_{conf}$ parameter that was 6.05 SD from the group mean (or 485.74 SD from the group mean when fit without them) which hence skewed the group-level fit. We analysed the posterior distributions using 89% credible intervals, following the suggestion that these are more stable than 95% intervals for analysing Bayesian posterior distributions[64]. Details of the model implementation in

STAN, the model fitting procedure, and the model simplification used can be found in Supplementary Information.

**Fitting internal noise and decision bias.** To measure the internal noise ($\sigma_{prior}$ and $\sigma_{likelihood}$) as well as the decision bias for each participant, we used the approach taken by Lisi et al.[29], and adapted the scripts available at https://osf.io/w74cn/. We assumed $\sigma_{prior}$ and $\sigma_{likelihood}$ to be the same, as we used the same stimuli for leads and targets. We fit four different psychometric functions to participants' decisions in the control task, as well as the first decision of each main task trial in Exp. 1 and 2, as these decisions all took place without informative priors (50/50 chance of right vs left). The four psychometric functions were (1) a simple function that includes only the internal noise as a free parameter, (2) a function with internal noise as well as decision bias, (3) a function with internal noise as well as a lapse term, and finally (4) a function that includes internal noise, decision bias, as well as lapse. The lapse term accounts for the possibility that participants might have made stimulus-independent lapses such as attentional or motor lapses. These four functions were fit and we then used the parameter values retrieved from taking an Akaike-weighted combination of the four estimates. For the modelling analysis, we transformed each participants' raw coherence values (coh) into units of their own internal noise. We additionally transformed their right versus leftward coherence values to take into account their own decision bias. Together, this transformation yielded the following definition of normalized stimulus strength $s$:

$$s = \frac{(\text{dir*coh}) - \text{bias}}{\sigma} \quad (1)$$

where dir is equal to −1 for leftward stimuli and +1 for rightward stimuli. This transformation allowed us to set the internal noise to 1 in all equations below.

**Flexible model.** The Flexible model includes three free parameters per participant—the prior weighting parameter at the decision level ($w_{choice}$), the prior weighting parameter at the confidence level ($w_{conf}$), and the confidence bias parameter ($b$). $w_{choice}$ quantifies the relative influence of the prior (compared to the likelihood) in the target decision. This influence of the prior can be captured computationally by shifting the decision criterion. Then, the probability of choosing right ($\Phi_{right}$) in the target decision is based on the probability that the perceived target stimulus was to the right of the shifted decision criterion. The decision criterion ($\theta$) is shifted proportionally to the weighted prior:

$$\theta = -\frac{|r_{lead}|}{w_{choice}} \quad (2)$$

where $r_{lead}$ is the internal response generated from the lead stimulus and internal noise. This shifting of the decision criterion is computationally equivalent to having a rightward prior equal to the decision confidence on the lead decision (Fig. 8a), but with the prior variance misestimated according to $w_{choice}$[29]. The likelihood of a rightward target decision was then computed, exactly as in the work of Lisi et al.[29], as:

$$\int_0^{+\infty} p\left(d_{target}^{right} | \theta = -\frac{r_{lead}}{w_{choice}}, s_{target}\right) p\left(r_{lead} | d_{lead}^{right}, s_{lead}\right) dr_{lead} \quad (3)$$

following a rightward lead decision, and as:

$$\int_{-\infty}^0 p\left(d_{target}^{right} | \theta = \frac{r_{lead}}{w_{choice}}, s_{target}\right) p\left(r_{lead} | d_{lead}^{left}, s_{lead}\right) dr_{lead} \quad (4)$$

following a leftward lead decision, where $s$ represents the stimulus and $d$ represents the decision. Because we as experimenters do not have access to the internal signals of the participant, the left term in the integral captures the probability of an internal target signal to the right of the shifted decision criterion – shifted according to the weighted prior signal, which derives from the lead stimulus. The right term weights this by the likelihood of that internal prior signal, given the lead stimulus, and these terms are marginalised across the possible prior signal values. Note that the model had to be adapted slightly to suit Experiment 3, which did not follow this dual-decision setup and hence did not have the prior come from a lead stimulus. Details of the necessary changes can be found in Supplementary Information (Section 'Flexible model–Experiment 3').

Confidence was then modelled as the perceived posterior probability of being correct, combining the prior and likelihood (Fig. 8a). The relative influence of the prior on confidence is captured by $w_{conf}$. This weighted rightward prior in confidence ($p(R)_{conf}$) is equal to the decision confidence from the lead decision, with the variance of the prior misestimated according to $w_{conf}$:

$$p(R)_{conf} = \Phi\left(\frac{|r_{lead}|}{b * w_{conf}}\right) \quad (5)$$

The strength of the likelihood depends on the incoming information from the target stimulus, and was defined as the likelihood of having gotten the internal target signal ($r_{target}$) if there had been a rightward target stimulus ($R$, or $s_{target} > 0$):

$$p(r_{target}|R) = \Phi\left(\frac{r_{target}}{b}\right) \quad (6)$$

Confidence bias, $b$, acts similarly to the weighting parameters, capturing a misestimate of the variance of information. However, unlike the weighting parameters (which only act on the prior), b captures an equal misestimation of both the prior variance and likelihood variance, and therefore reflects an overall tendency to interpret stimulus information as being either noisier or clearer than it really was. $b > 1$ reflects a general overestimation of the signal variance, and scales confidence in both choice alternatives towards 50%. $b < 1$ reflects a general underestimation of the signal variance, and scales confidence in the chosen alternative towards 100% and the rejected alternative towards 0%. While this still captures a general over- or underconfidence, it is different from a report bias, which would just freely shift the final confidence report. This was chosen because we wanted a confidence bias that was more perceptual in nature, and less flexible compared to a report bias, which would overpower the weighting parameters.

The posterior combined the prior and likelihood according to Bayes rule, and confidence in a rightward choice was then computed as:

$$conf_{right} = \frac{p(R)_{conf}\, p(r_{target}|R)}{(1 - p(R)_{conf})(1 - p(r_{target}|R)) + p(R)_{conf}\, p(r_{target}|R)} \quad (7)$$

Confidence in a leftward choice was then equal to (1 - $conf_{right}$). Similarly to $w_{choice}$, $w_{conf}$ captured the prior strength that would account for each confidence rating by scaling the variance of the internal prior signal relative to the internal likelihood signal. By implementing these two weighting parameters in the same way, we could then directly compare them. The Flexible model was the only model in which the weighting of the prior information was allowed to differ between discrimination decisions and confidence ratings.

**Flat prior model.** The Flat Prior model captured decisions and confidence in the same way as the Flexible model except that the prior

information had no influence on the target decision or confidence, so the lead and target decisions were modelled as independent and confidence was modelled as the decision confidence about only the target stimulus. Computationally this meant forcing the prior for a rightward target stimulus to be equal to 0.5, or an uninformative prior, which was analogous to setting $w_{choice}$ and $w_{conf}$ to be infinitely large. The only free parameter in this model was the confidence bias ($b$).

**Optimal model.** The Optimal model only differed from the Flexible model in that it assumed the prior information to be optimally precision-weighted relative to the likelihood. This meant that $w_{choice}$ and $w_{conf}$ were both equal to 1, with only $b$ as a free parameter.

**Equal model.** The Equal model was also the same as the Flexible model except that the use of the prior information was assumed to be the same in decisions and confidence, although it could stray from optimal. Computationally, this meant that $w_{choice}$ and $w_{conf}$ were forced to be equal to one another, and so only one weighting parameter ($w$) was fit, which was then used as both $w_{choice}$ and $w_{conf}$. Again, $b$ was still fit to capture an overall confidence bias in this model.

**Model comparison.** We compared the ability of our four models to account for the behavioural data of the remaining 20 participants after removing the outlier participant. To do this, we performed a 10-fold LOGO-CV, in which we left out 2 participants at a time, fit each model to the remaining 18 participants, and then measured the predictive performance of those fit models for predicting the data of the left-out participants using the 'loo' package[65]. The log predictive density for each model for each fold was stored and we then computed the overall expected log pointwise predictive density for each model, and compared them. We considered a model to fit the data better if the magnitude of the difference in expected log pointwise predictive density (elpd_diff) was at least 4, and at least 2 times larger than the standard error of the difference (se_diff)[66].

### Reporting summary
Further information on research design is available in the Nature Portfolio Reporting Summary linked to this article.

## Data availability
The raw experimental data generated in these experiments have been deposited in a public repository[67] on Zenodo under https://doi.org/10.5281/zenodo.8131976. Source data are provided with this paper.

## Code availability
Reproducible analysis scripts and models are publicly available under https://gitlab.com/MarikaConstant/priors-in-confidence.

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

## Acknowledgements
We thank Martin Krueck for help with an earlier version of the modelling presented here. We thank Matteo Lisi for suggesting that we explore an alternative explanation of our results, and for the code to do so. M.C.'s work was funded by the Deutsche Forschungsgemeinschaft (DFG, German Research Foundation)—337619223 / RTG2386. M.C. and E.F. were supported by a Freigeist Fellowship from the Volkswagen Foundation (grant number 91620) to E.F.. M.P. was supported by a Postdoc. Mobility fellowship from the Swiss National Science Foundation (P400PM_199251) to M.P.. N.F. was supported by the European Research Council (ERC) under the European Union's Horizon 2020 research and innovation programme (grant agreement no. 803122 to N.F.). The funders had no role in the conceptualization, design, data collection, analysis, decision to publish, or preparation of the manuscript.

## Author contributions
M.C. and E.F. designed the experiments and prepared the manuscript. M.C. collected the data. M.C. and M.P. developed the models, and M.C. performed the data analysis. M.C., M.P., N.F., and E.F. contributed to interpretation of the results and to the editing of the manuscript.

## Funding

## Competing interests
The authors declare no competing interests.
