## [Peer Review File · Nature Communications]

Prior information differentially affects discrimination decisions and subjective confidence reportsREVIEWER COMMENTS

Reviewer #1 (Remarks to the Author):

Constant and colleagues report the results of two studies investigating how prior information is used in the act of making a decisions and computing confidence in the accuracy of that decision. The paper addresses a highly interesting question, it is very well written and was therefore a pleasure to read. I think this is great work that definitely advances our understanding of the computations underlying decision confidence. At the same time, I do have rather major concerns regarding interpretation, the quality of model fit, an alternative explanation and the results of experiment 2.

Major comments

The idea of creating matched pairs of posterior information is excellent and very innovative. At the same time, throughout the manuscript I was waiting for some proof that this is indeed the case. So far, this argument remains very intuitive. Most importantly, as I understand posterior information is matched _conditional_ on confidence being a perfect read-out of posterior uncertainty. Could it be that if confidence is a noisy read-out of posterior uncertainty, it is no longer the case that posterior information is matched (i.e. and so then perhaps people actually de optimally make use of their - noisy- prior information?). There is quite some evidence out there for metacognitive noise (e.g., Shekhar & Rahnev, 2021, Psych Rev; Guggenmos, 2022, eLife; etc.), so could this be a reasonable alternative explanation of the findings?

I am generally a bit concerned about the model fit. For both experiments, the model fits the accuracy data well but it does not do a good job with confidence. In fact, it often predicts a full .1 difference with the empirical data; which is not a small deviation on a .5 scale. I was not very satisfied with the explanation for this (simplification in the modeling), and rather worried about the interpretation of wconf given the far from ideal model fit. Can we really trust these values? This is even more problematic in Experiment 2, where the condition effect seems to be qualitatively different in the data vs model (but no stats are provided, so hard to judge).

I applaud the authors for replicating their own work in a pre-registered replication. At the same time, I wondered why the results for confidence in Experiment 2 were not mentioned in the main text (nor in the supps, were there was only one sentence mentioning a slightly different result from that in Experiment 1, but without reporting any stats). Please provide a full description of the results in the manuscript.

Minor comments

The mixed models, reported in Table 1, are rather complex and so it does not come as a surprise that these do not convergence with such relatively low sample size.

It only became clear halfway the manuscript that priors are underused for both choices and confidence. This mostly caused by sentences such as e.g. in the abstract "priors were underweighted in discrimination decisions, but used to a greater extent in confidence about those decisions" which can (more easily) be interpreted as saying that priors are overweighted for confidence (rather than being less underweighted).

The fact that the task was "gamified" is mentioned very prominently (i.e. in the abstract), but it is unclear why this the case. Participants just make two random dot motion decisions which are connected to each other. I think it is great to have some background story for participants, but putting so much emphasis on this aspect (e.g. in the abstract) seems to detract a bit from the actual findings

On line 297, rather than merely stating that the empirical pattern can be explained in this way, it would be useful to provide the reader with an intuition as to why the empirical pattern in confidence follows from underusing priors.

The folded-X pattern is not the best evidence for the claim that confidence is Bayesian, to my knowledge most models of confidence predict this pattern.

Reviewer #2 (Remarks to the Author):

The authors use a clever manipulation to evaluate whether priors or likelihoods are used similarly across decisions and metacognitive confidence judgments in those decisions. This approach revealed that priors and likelihoods are not used equivalently across decisions and confidence judgments: specifically, priors are underweighted in discrimination decisions relative to their weight contributing to confidence judgments.

This work is timely and highly relevant to the rapid pace at which we are investigating how decision confidence is computed in neural circuitry. The manuscript is clearly written and easy to follow. I have some comments that I hope will help maximize the impact of the paper by exploring some potential alternative explanations for the results and showing how they relate to other ongoing discussions in the field.

Specific comments:

- Were there cases where the target stimulus was completely uninformative, or going in the opposite direction of the prior established by the lead stimulus? That is, was the deterministic structure of the target stimulus direction always 100%? I ask this because catch trials where the target was incongruent with the expected direction based on response to the lead stimulus could help explore the nuanced nature of the effect, namely cases where prior expectation was strongly INCONGRUENT with incoming likelihood information (strong or weak). In looking at the methods, it appears such catch trials were not included. Could the authors discuss why this choice was made, and speculate (perhaps based on their models) what might have happened if they had been?

- Perhaps I missed this, but specifically how was metacognitive inefficiency taken into account in model fitting? Specifically, the authors cleverly induce a 'stronger' or 'weaker' directional prior via manipulating the coherence of the lead stimulus; however, the exact value of this prior would also depend on the observer's performance on this lead stimulus *and* the observer's metacognitive efficiency. How was this taken into account? This could effectively show up as underweighting of the prior if the observer were particularly metacognitively inefficient (i.e., the observer built a 'less strong' prior than they should have based on their accuracy on the lead stimulus). Moreover, metacognitive inefficiency could interact with the Bayesian model of decisions (lines 229+) in which the assumption appears to be consistent over- or consistent under-estimation of the precision of the prior, rather than the presence of metacognitive noise simply corrupting this (overall potentially unbiased) estimation procedure. This analysis and its results should be included in the manuscript.

- Is it possible that the apparent discrepancy, in that confidence underweights the prior LESS than decisions do, could be due to metacognitive inefficiencies? That is, is it the case that confidence optimally uses the (inefficiently estimated) prior at the noise ceiling due to confidence leak from lead stimulus to target stimulus? I am not sure I am being clear here (and not sure exactly how this would work), but it seems there might be some effect of 'confidence leak' between lead and target stimulus that could interact with the measurement of how effectively the prior is used. There also feels like a conceptual differentiation here, which is that the 'kind' of information resulting from estimating the prior from the lead stimulus (i.e., decision confidence) is the same 'kind' of information used in the confidence judgment in the target stimulus. Thus, perhaps it's not so much that the prior is underweighted, but that the use of the prior is corrupted when translating between type 2 and type 1 reports. Much like we often assume that type 2 noise corrupts signals when they flow from the type 1

to the type 2 space, here the authors are asking whether information 'flowing' from the type 2 to the type 1 space can be used optimally; if the same kind of 'translation' noise is corrupting the information regardless of whether it goes from type 1 → type 2 or type 2 → type 1, then there might again be some relationship between underweighting of prior information in the decision and metacognitive efficiency (M-ratio), as discussed in the previous bullet. Perhaps the authors could consider and discuss these issues.

- There is a lot of work on the "right" kind of metacognitive noise/corruption: normally-distributed type 2 noise, multiplicative internal noise, log-normally distributed internal noise, use of decision-congruent evidence while downweighting/ignoring decision-incongruent evidence... basically, as the authors I know are aware, this field is a bit of a mess right now. Nevertheless, it feels important to discuss the potential implications of these other kinds of noise/metacognitive inefficiency/estimation assumptions on the results. The model fitted has an internal noise factor fitted to each subject (lines 305-306, for example)

Reviewer #3 (Remarks to the Author):

In this manuscript, Constant and colleagues report on empirical and modelling investigations of decisions and metacognitive judgments. Using an elegant dual-decision task, adapted from previous work, they independently manipulate the strength of priors and likelihoods on which decisions and confidence judgments are based. In two experiments (N = 21 + 25), they report that priors are underweighted in decisions, but used to a greater extent in confidence judgments about those decisions. They finally proceed to account for this result with a Bayesian model.

There is a lot to like in the manuscript: the task is very elegant, it is combined with straightforward hypotheses, that are tested thanks to a clear and sharp operationalization (experimental manipulation). The results seem robust, as they replicate in the 2 (close to identical) instantiations of the task.

My potential enthusiasm is nonetheless tempered by several conceptual and methodological concerns which should be addressed before I can recommend the manuscript for publications.

Main concerns

1. First, the manuscript is written in a very sharp/straightforward way. This makes it a pleasure to read but comes at the cost of oversimplifications and overgeneralization. In particular, although I find the task very elegant, I cannot help but worry that some (if not most) of the main conclusions/findings are very specific/idiosyncratic of the present experimental setup (dual-task), where the prior is defined by the performance of a first decision task. It seems to me that if one just wanted to test the weight of priors vs likelihood weights in decision and confidence judgments, another (more?) straightforward way would be a single-task that includes an exogenous cue, which provides an exogenous (rather than endogenous) prior evidence (as per instruction, e.g. a dot whose color correlate with probability of the true state of the world being A/B). This brings the question: what is specific vs generalizable in the current findings, in this dual task? Of course, it is not my role to impose a new experiments (though it would be very informative). But I think the authors should, at the very least, 1) better delineate the specificity of the current setup (dual task) and how this can help the reader better interpret/conceptualize the results, and 2) be more explicit about the (potential) specificity of their results to the current setup.

2. Confidence judgments seem, overall, very in-sensitive to the overall amount of (posterior) evidence (Figure 2C, 6B). Would a model without main effect of condition (stronger Lead/Target) still replicate the commonly found effect of choice accuracy and interaction between accuracy and evidence on confidence (so called X-pattern)? Also, I'm wondering to what extent it would make sense to reproduce the panel of Figure 2B and 2C (accuracy and confidence) as a function of prior strength and

likelihood strength (rather than just posterior strength), to provide new dimensions to evaluate pattern of results to validate/falsify the tested models.

3. Although a normative model is always an excellent benchmark, I am actually a bit frustrated that the authors did not explore other ways to explain their findings. For instance, it could very well be that there are non-additive interactions between the two-steps of dual task. Notably, decision-makers are known to behave in a confirmatory/motivated way: basically, they process evidence, make decisions and evaluate confidence in a way that is confirmatory see e.g. (Rollwage et al., 2020; Talluri et al., 2018, 2021) – this actually also connects/extends to the topic of choice-congruent evidence mentioned by the authors in the introduction. I'm therefore wondering whether such a process could happen here, where the target-likelihood could be evaluated in an asymmetric way, to confirm the lead-choice (creating an asymmetry between confidence in correct versus incorrect, that also depends on the condition lead/target first). This may be a direction to explore, notably to explain one of the current model main misfit, which is a systematic underestimation of confidence judgments (Figures 4B, 6B)

Other

4. Regarding model recovery (Figure S4), I encourage the authors to commit to a more classic approach: define a model-comparison winning criteria, and produce proper confusion matrices.

5. This might seem completely trivial, but the fact that step-1 correct choices are deterministically associated with the right direction as the step-2 creates could be somewhat "dangerous", if there is any decision bias toward one side. Why not balancing this (even if just between-subject)?

Rollwage, M., Loosen, A., Hauser, T. U., Moran, R., Dolan, R. J., & Fleming, S. M. (2020). Confidence drives a neural confirmation bias. *Nature Communications*, 11(1), Article 1.

<https://doi.org/10.1038/s41467-020-16278-6>

Talluri, B. C., Urai, A. E., Bronfman, Z. Z., Brezis, N., Tsetsos, K., Usher, M., & Donner, T. H. (2021). Choices change the temporal weighting of decision evidence. *Journal of Neurophysiology*, 125(4), 1468–1481. <https://doi.org/10.1152/jn.00462.2020>

Talluri, B. C., Urai, A. E., Tsetsos, K., Usher, M., & Donner, T. H. (2018). Confirmation Bias through Selective Overweighting of Choice-Consistent Evidence. *Current Biology*, 28(19), 3128-3135.e8. <https://doi.org/10.1016/j.cub.2018.07.052>

REVIEWER COMMENTS

Reviewer #1 (Remarks to the Author):

Constant and colleagues report the results of two studies investigating how prior information is used in the act of making a decisions and computing confidence in the accuracy of that decision. The paper addresses a highly interesting question, it is very well written and was therefore a pleasure to read. I think this is great work that definitely advances our understanding of the computations underlying decision confidence. At the same time, I do have rather major concerns regarding interpretation, the quality of model fit, an alternative explanation and the results of experiment 2.

Major comments

1.1. The idea of creating matched pairs of posterior information is excellent and very innovative. At the same time, throughout the manuscript I was waiting for some proof that this is indeed the case. So far, this argument remains very intuitive.

We thank the reviewer for noting the quality and originality of our work. We agree that it is important to consider whether the conditions are truly matched in terms of available internal information.

Indeed it is in principle possible that the internal information is not perfectly matched across conditions, despite the externally available posterior information being matched, and despite our efforts to minimize this risk by having both the prior and likelihood come from the same 'kind' of information, namely a dot motion stimulus. Importantly however, our model is agnostic about the matched conditions, and just treats trials as sequences of signal strengths (a prior and then a likelihood). The model, therefore, does not rely on the assumption of matched internal information.

While not important for the model, including the two conditions in the design allowed us to conduct model-free analyses of the behavioural data. These revealed converging evidence with the modeling. The consistent results we found between the two analyses support the validity of our conclusions. We clarified this point in the revised manuscript (pg. 8), copied here:

“Additionally, the models we fit to the data do not rely on the matched pairs of posterior information, and can hence provide important converging lines of evidence to validate the model-free results in case the conditions were not perfectly matched in their internal processing.”

In fact, since the model does not rely on the matched conditions, it is possible to test the same hypothesis without creating matched conditions at all, which we now do in a new third experiment, and find the same results.

Finally we note that our main conclusion on the asymmetry between the impact of priors on decisions and confidence does not rely on the differences between conditions

(Stronger-Lead vs Stronger-Target), but rather on the differences between target discrimination decisions and confidence ratings. Even if there were a systematic difference between the two conditions, this would have affected the internal information available for both target discrimination decisions and in confidence equally. As a result, the asymmetry between these two processing levels can be safely interpreted.

Most importantly, as I understand posterior information is matched _conditional_ on confidence being a perfect read-out of posterior uncertainty. Could it be that if confidence is a noisy read-out of posterior uncertainty, it is no longer the case that posterior information is matched (i.e. and so then perhaps people actually do optimally make use of their -noisy- prior information?). There is quite some evidence out there for metacognitive noise (e.g., Shekhar & Rahnev, 2021, Psych Rev; Guggenmos, 2022, eLife; etc.), so could this be a reasonable alternative explanation of the findings?

It is true that the internal confidence computation that forms the prior is impacted by metacognitive noise, but this has no effect on our conclusions, even from the model-free analyses. This is because the experimental conditions (Stronger-Lead and Stronger-Target) are matched in expected target decision accuracy and mean target confidence even in the presence of metacognitive noise. We now discuss this important point in the manuscript on pg. 21 in a section on alternative explanations, as well as in depth in Supplementary Materials:

“In the dual-decision paradigm, the rightward prior is susceptible to metacognitive noise, because it is itself based on an internal confidence computation. The strength of the rightward prior is equal to the point estimate of confidence on each trial. So, just as metacognitive noise makes confidence judgments noisier across trials, it will lead to more variable estimated prior strengths across trials. In other words, the priors will sometimes be estimated as stronger or weaker than appropriate, making them less reflective of the correct priors. Note that in this task, the effect that metacognitive noise has on priors is different from the intuition, stemming from Bayesian integration, that noisier signal distributions at the metacognitive level will make the prior weaker overall. This is because the strength of the prior is based on the point estimate of confidence on a given trial, rather than the width of the Type 2 distributions. So, this poorly estimated prior will make target decision accuracy worse, but will not lead to consistently over- or underestimated priors, or to differences between conditions.”

We now also include Figure 8B to further clarify how metacognitive noise interacts with our generative model, copied in Response 2.2 below.

Moreover, we agree with the reviewers on the importance of considering the role of metacognitive noise in our model, and we have now included a thorough analysis of different possible implementations of metacognitive noise in Supplementary Materials, with the figure summarizing these analyses copied in Response 2.2 below. Additionally, Exp. 3 now demonstrates that the same results hold even when the prior is an

exogenous probabilistic cue, and is therefore no longer susceptible to metacognitive noise.

1.2. I am generally a bit concerned about the model fit. For both experiments, the model fits the accuracy data well but it does not do a good job with confidence. In fact, it often predicts a full .1 difference with the empirical data; which is not a small deviation on a .5 scale. I was not very satisfied with the explanation for this (simplification in the modeling), and rather worried about the interpretation of w_{conf} given the far from ideal model fit. Can we really trust these values? This is even more problematic in Experiment 2, where the condition effect seems to be qualitatively different in the data vs model (but no stats are provided, so hard to judge).

The model simplification was made in order to reduce complexity issues that limited our capacity to fit the model hierarchically, and also to help address overfitting concerns. This complexity comes because the non-simplified model includes two free parameters for the internal signal strengths of each trial. However, the simplification comes at a cost: The model fits are less flexible, but more importantly, the model estimates are skewed in predictable ways, tending to underestimate confidence.

Below, we respond to the reviewer's concerns in three ways. In (a) we explain the predictable ways in which the simplification affects the parameter estimates, and why the fit parameters can be trusted. Then (b), we explain more in depth how and why the simplified model underestimates confidence, and include this in the revised manuscript. Finally (c), we now fit the non-simplified model to each participant individually, providing better fits for confidence and further confirming the interpretation of the simplified model. We included these results as Supplementary Materials in the revised manuscript.

(a) Confidence was indeed systematically underestimated by the simplified model, but this does not impact the estimated differences between conditions, on which we based our interpretations. In order to ensure that this is true, we ran a parameter recovery analysis: We simulated data from the non-simplified model (sampling internal signals for use in the decision and confidence computations) and then fit the simplified model to that data. The results of this analysis can be seen in Figure S2 in the initial submission. This confirmed that the simplified model had no effect on w_{choice} , and — as we expected — the strongest effect on the confidence bias parameter, causing the general underestimation of confidence (see (b) below). The simplified model also resulted in greater values of w_{conf} (more underweighted confidence), though this effect was minor. Note that this goes in the opposite direction of our conclusions. Hence, if anything, fitting the simplified model made our interpretations more conservative.

(b) The simplified model bases confidence computations on the external signal strengths (or the means of the internal distributions), without sampling internal signals. In other words, internal signals are deterministic, given the stimulus shown. That means that,

for the simplified model, stimuli to the e.g. right will never lead to an internal sample that is on the “left” side of the decision criterion. As a consequence, the models’ confidence estimate for incorrect choices will be under 50%. Because ratings below 50% were not possible on our scale, the model must adjust its confidence bias parameter to increase confidence in incorrect trials, in order to get the predictions (consistently under 50%) to reach participants’ ratings (consistently above 50%).

Note that in our model the confidence bias parameter, b does not simply shift confidence for all conditions alike. This implementation would have overpowered the weighting parameters. Instead, we opted to implement a parameter that would more accurately describe confidence bias that was more perceptual in nature, and less flexible compared to a report bias. We now describe in depth in the manuscript how b is implemented (pg. 31):

“ b captures an equal misestimation of both the prior variance and likelihood variance, and therefore reflects an overall tendency to interpret stimulus information as being either noisier or clearer than it really was. $b > 1$ reflects a general overestimation of the signal variance, and scales confidence in both choice alternatives towards 50%. $b < 1$ reflects a general underestimation of the signal variance, and scales confidence in the chosen alternative towards 100% and the rejected alternative towards 0%. While this still captures a general over- or underconfidence, it is different from a report bias, which would just freely shift the final confidence report.”

Therefore, in order to drive predicted confidence in incorrect trials (below 50%) upwards towards 50%, the model must drive confidence in both alternatives towards 50% by increasing b . This in turn decreases confidence predictions in correct trials, as they too are pushed towards 50%. So, if b is high, then predicted confidence in correct trials is too low, but if b is low, then predicted confidence in incorrect trials is too low. This tradeoff forces an underestimation of confidence overall. This point is now explained in more depth on pg. 10-11 of the revised manuscript. In order to mitigate these limitations and explore them further, we fit another version of the model (‘Bounded Confidence Model’, see Figure S3) which forces confidence to be at least 50% in the fitting, to avoid skewing the confidence bias parameter. This indeed decreased the skewing, and our pattern of results remained, but unfortunately there were convergence issues with that model so we used it only as a further sanity check to ensure these limitations were not an issue for our interpretations. Together, we argue that the simplification made here limits interpretation less than the issues with model complexity and overfitting.

- (c) *Nonetheless, to further address the concerns about the model simplification, we now also fit the non-simplified model individually to each participant from Exp. 1, rather than hierarchically. This avoids the complexity constraints, because the number of free parameters scales with the total trial number. With this approach we found the same pattern of results ($w_{\text{choice}} > w_{\text{conf}}$) in 15 out of the 20 participants included in the*

hierarchical modeling, shown in the figure below, supporting our interpretations of the hierarchical model.

Also, we have included here (and in Supplementary Figure S4) the plots from sampling the subjectwise fits of that model, to demonstrate that the non-simplified model can fit confidence well. This also further highlights that it is the simplification that leads to the underestimation of confidence, rather than another feature of our model.

In sum, we show that even if the simplification might lead to different values of the model parameters, this does not change the conclusions. If anything, the simplification led to more conservative results than the non-simplified model.

1.3. I applaud the authors for replicating their own work in a pre-registered replication. At the same time, I wondered why the results for confidence in Experiment 2 were not mentioned in the main text (nor in the supps, were there was only one sentence mentioning a slightly different result from that in Experiment 1, but without reporting any stats). Please provide a full description of the results in the manuscript.

We had originally only included the statistics for the decision level results, since the focus of Exp. 2 was to demonstrate that adding more evidence accumulation time would not increase the use of the prior in discrimination decisions.

We now report the statistics for the confidence level results of Exp. 2 as well on pg. 15, but also call for caution in interpreting them: due to the finding of strongly underweighted prior in decisions, the behavioural predictions at the confidence level and hence the interpretation of these results become less clear.

Minor comments

1.4. The mixed models, reported in Table 1, are rather complex and so it does not come as a surprise that these do not convergence with such relatively low sample size.

Following Barr and colleagues (2013), we used the approach of selecting the most complex random effects structure that converged on the full model. We want to clarify that the mixed models reported in Table 1 are the models that did converge.

*Barr, D. J., Levy, R., Scheepers, C., & Tily, H. J. (2013). Random effects structure for confirmatory hypothesis testing: Keep it maximal. *Journal of memory and language*, 68(3), 255-278.*

1.5. It only became clear halfway the manuscript that priors are underused for both choices and confidence. This mostly caused by sentences such as e.g. in the abstract “priors were underweighted in discrimination decisions, but used to a greater extent in confidence about those decisions” which can (more easily) be interpreted as saying that priors are overweighted for confidence (rather than being less underweighted).

We agree that our wording may have been confusing. We have adjusted this phrasing to make this clearer from earlier on in the manuscript.

1.6. The fact that the task was “gamified” is mentioned very prominently (i.e. in the abstract), but it is unclear why this the case. Participants just make two random dot motion decisions which are connected to each other. I think it is great to have some background story for participants, but putting so much emphasis on this aspect (e.g. in the abstract) seems to detract a bit from the actual findings

In order to take the gamification out of focus, we have removed mention of it in the Abstract and Introduction, and have only left it when there was a description of the methods – once in the main text when giving the task description and figure legend, and in the Methods section.

1.7. On line 297, rather than merely stating that the empirical pattern can be explained in this way, it would be useful to provide the reader with an intuition as to why the empirical pattern in confidence follows from underusing priors.

We have now framed this prediction with more explanation (pg. 10).

1.8. The folded-X pattern is not the best evidence for the claim that confidence is Bayesian, to my knowledge most models of confidence predict this pattern.

We agree and have now noted this in the revised manuscript. We now simply state on pg. 1 that: "This Bayesian confidence model has been highly influential, and it has been tested and supported empirically, in both animals and humans⁸⁻¹⁴, though alternatives have been proposed¹⁵⁻¹⁷."

Reviewer #2 (Remarks to the Author):

The authors use a clever manipulation to evaluate whether priors or likelihoods are used similarly across decisions and metacognitive confidence judgments in those decisions. This approach revealed that priors and likelihoods are not used equivalently across decisions and confidence judgments: specifically, priors are underweighted in discrimination decisions relative to their weight contributing to confidence judgments.

This work is timely and highly relevant to the rapid pace at which we are investigating how decision confidence is computed in neural circuitry. The manuscript is clearly written and easy to follow. I have some comments that I hope will help maximize the impact of the paper by exploring some potential alternative explanations for the results and showing how they relate to other ongoing discussions in the field.

Specific comments:

2.1. - Were there cases where the target stimulus was completely uninformative, or going in the opposite direction of the prior established by the lead stimulus? That is, was the deterministic structure of the target stimulus direction always 100%? I ask this because catch trials where the target was incongruent with the expected direction based on response to the lead stimulus could help explore the nuanced nature of the effect, namely cases where prior expectation was strongly INCONGRUENT with incoming likelihood information (strong or weak). In looking at the methods, it appears such catch trials were not included. Could the authors discuss why this choice was made, and speculate (perhaps based on their models) what might have happened if they had been?

Indeed we did not include catch trials in which the deterministic rule did not hold. This was because the prior was formed based on an internal confidence value which was already quite subtle for participants to learn, so we were concerned that including trials in which this was false would interfere with their understanding of the task.

We agree however that looking at trials in which the prior was strongly incongruent with the likelihood is interesting. The leftward target stimuli offered us the opportunity to do so as explained below. We now added the following analysis to the Supplementary Materials:

“We assume that confidence ranged from 50-100%, and hence the rightward prior also ranged from 50-100%. This means that leftward target stimuli would typically be incongruent with the prior. We can then examine this subset of prior-incongruent trials, which were trials with leftward target stimuli, or in other words, with incorrect lead decisions. Out of these prior-incongruent trials, the strongest incongruence would occur on trials with the strongest rightward prior, which occur with high lead decision confidence, despite the incorrect lead decision. Though it is difficult to identify incorrect lead decisions that have high confidence, since we do not have explicit lead confidence ratings, the folded-X pattern of the Bayesian confidence model predicts these to occur at the lowest stimulus intensity. Together, we assume the prior-incongruent trials to be ones with incorrect lead decisions, and we expect stronger incongruency (and therefore lower accuracy) with lower lead stimulus intensities. This is also shown in model predictions in Figure S11B, with lower target decision accuracy for lower lead stimulus intensities for prior-incongruent trials. We find the data to go in line with these predictions (Figure S11A).

Figure S11. Target Accuracy on Prior-Incongruent vs Prior-Congruent Trials. A. Observed Target Accuracy. Observed target decision accuracies per lead stimulus level at each target stimulus level, which are split into panels. Accuracies are shown based on prior-congruent (blue) versus prior-incongruent trials (red), with prior-incongruent trials referring to leftward target stimuli, or incorrect lead decision trials. Error bars capture standard deviation of accuracies across participants. **B. Simulated Target Accuracy with Optimal Prior Weighting.** Simulated target decision accuracies given optimal weighting of the prior ($w_{choice} = 1$), shown based on prior-congruent (blue) versus prior-incongruent trials (red). **C. Simulated Target Accuracy with Underweighting of Priors.** Simulated target decision accuracies given optimal weighting of the prior ($w_{choice} = 2$), shown based on prior-congruent (blue) versus prior-incongruent trials (red).

Further, we qualitatively compared the data to model predictions on prior-congruent versus prior-incongruent trials, given optimal weighting and underweighting of the prior. On prior-congruent trials, participants did not perform as well as would be predicted of an optimal observer (Figure S11B, blue), suggesting that they are underusing the informative prior on those trials, in line with what is shown in the “Underweighting” model simulations (Figure S11C, blue). However, on prior-incongruent trials, participants seem to ‘stick to’ the rightward prior that they form, although this then goes against the likelihood, which actually keeps their accuracy low, close to that of the “Optimal” model for incongruent trials (Figure S11B, red). Interestingly, this might suggest that the

underweighting of the prior in target decisions that we find is driven more by prior-congruent trials underusing the prior, rather than by prior-incongruent trials. We then consider confidence predictions following prior-congruent versus prior-incongruent target stimuli. With prior-incongruent target stimuli (aka leftward target stimuli), correct target decisions were prior-incongruent target decisions (aka they chose “left”). Incorrect target decisions were prior-congruent target decisions (aka they chose “right” despite the leftward stimulus). This means that when the target stimulus is very weak and participants are presumably unsure about the target stimulus itself, confidence is actually expected to be higher following incorrect decisions, since those go in line with the prior (Fig. S12B, left panel, red). We found target confidence to follow this pattern for incongruent trials (Fig. S12A, red). However, we found that pattern for all levels of the target stimulus, not just the low precision level. To investigate what this suggests about the prior weighting, we simulated data with both optimal prior weighting and with prior weighting that is stronger in confidence than in decisions, in line with our other results, and explored confidence patterns for prior-incongruent versus prior-congruent trials. This reveals that stronger weighting of the prior in confidence compared to decisions can cause what we see in the data for prior-incongruent trials: higher confidence following incorrect decisions on prior-incongruent trials, at all levels of target stimuli used (Fig. S12C, red). This adds additional evidence to support our conclusion of stronger use of priors in confidence compared to decisions.”

A. Data

B. Model - Optimal Weighting

C. Model - $w_{choice} > w_{conf}$

Figure S12. Confidence on Prior-Incongruent vs Prior-Congruent Trials. A. Observed Confidence.

Observed mean confidence following correct and incorrect trials, shown per lead stimulus level at each target stimulus level, which are split into panels. Confidence is also divided by prior-congruent trials (blue) and prior-incongruent trials (red). For prior-incongruent trials, note that correct decisions involve decisions that are in line with the likelihood but not the prior, and incorrect decisions involve decisions that are in line with the prior but not likelihood. Error bars capture standard deviation of mean confidence across participants. **B. Simulated Confidence with Optimal Prior Weighting.** Confidence patterns for prior-congruent versus prior-incongruent trials, from simulated data with optimal prior weighting (w_{choice} and $w_{conf} = 1$). **C. Simulated Confidence with Stronger Prior Weighting in Confidence than in Decisions.** Confidence patterns for prior-congruent versus prior-incongruent trials, from simulated data with stronger prior weighting in confidence compared to decisions ($w_{conf} = 1, w_{choice} = 2$).

We thank Dr. Peters for pointing us to this more nuanced exploration of our results.

2.2. - Perhaps I missed this, but specifically how was metacognitive inefficiency taken into account in model fitting? Specifically, the authors cleverly induce a ‘stronger’ or ‘weaker’ directional prior via manipulating the coherence of the lead stimulus; however, the exact value of this prior would also depend on the observer’s performance on this lead stimulus *and* the observer’s metacognitive efficiency. How was this taken into account? This could effectively show up as underweighting of the prior if the observer

were particularly metacognitively inefficient (i.e., the observer built a ‘less strong’ prior than they should have based on their accuracy on the lead stimulus).

*The presence of metacognitive noise will make the Type 2 distributions noisier than the Type 1 distributions. Because the prior here was computed at the metacognitive level, there is a natural intuition that the prior will be weakened by metacognitive noise, since the distributions get wider. However, critically, this is **not** in line with the setup of our paradigm. Here, the strength of the prior is equal to the point estimate of confidence on each trial. It is not equal to the width of the Type 2 signal distributions. Hence, just as metacognitive noise jitters confidence estimates across trials and makes them less reflective of accuracy, but not biased, metacognitive noise will jitter the estimated strength of the prior across trials and make it less reflective of the true prior, but not systematically weakened. To clarify this unintuitive and important point, we now include an additional panel in Figure 8 in Methods (copied below), capturing how metacognitive noise acts in our generative model, making the estimated prior noisier across trials.*

Moreover, metacognitive inefficiency could interact with the Bayesian model of decisions (lines 229+) in which the assumption appears to be consistent over- or consistent under-estimation of the precision of the prior, rather than the presence of metacognitive noise simply corrupting this (overall potentially unbiased) estimation procedure.

Similarly to what we described above, it is true that noisier prior estimation due to metacognitive noise will decrease target decision accuracy, because the estimated prior will deviate from the true prior. But, critically, this deviation will lead to noisier, not biased, prior estimates across trials. Hence, this will not interact with our conditions and it cannot explain our results, namely a consistent underweighting of the prior in target decisions leading to lower accuracy in the Stronger-Prior condition.

This analysis and its results should be included in the manuscript.

A discussion of the role of metacognitive noise in our design is now included in the manuscript (pg. 21 and 32, and Supplementary Materials pg. 8-10).

Also, we agree with all reviewers' suggestions to consider how metacognitive noise interacts with our model and results. Hence, we include in the Supplementary Materials a thorough investigation of different possible implementations of metacognitive noise and how they impact predictions, confirming that none of them can account for our results. The figure summarizing this analysis is copied here. Briefly, this analysis first shows that metacognitive noise cannot account for the decision-level difference between conditions (top row), and then shows that, given the underweighting of priors in decisions, metacognitive noise also cannot account for the asymmetrical confidence-level results (bottom row).

Figure S7. Metacognitive Noise Simulations. A. Observed Target Decision and Confidence Results. (Top) Decision level results showing target accuracy per condition and posterior information level. (Bottom) Confidence level results showing mean confidence following correct and incorrect trials per condition and posterior information level. **B. Simulations with No Metacognitive Noise.** (Top) Decision level results given optimal weighting of the prior ($w_{choice} = 1$), with no metacognitive noise. This is used as a comparison for C-E, to show the effect of adding metacognitive noise. (Bottom) Confidence level results given the decision level result found ($w_{choice} = 2.17$) but equal weighting of the prior in confidence ($w_{conf} = 2.17$), and no metacognitive noise. The fit confidence bias ($b = 2.18$) was also used. This is used as a comparison for C-E to show the effect of adding metacognitive noise. **C. Simulations with Balanced Metacognitive Noise.** (Top) Decision level results given optimal prior weighting but with metacognitive noise in estimating the precision of both the prior and likelihood. (Bottom) Confidence level results given equal underweighting of the prior between decisions and confidence, and added metacognitive noise in estimating the precision of both the prior and likelihood. **D. Simulations with Metacognitive Noise Corrupting Prior Only.** (Top) Decision level results given optimal prior weighting but with metacognitive noise in estimating just the prior. (Bottom) Confidence level results given equal underweighting of the prior between decisions and confidence, and added metacognitive noise in estimating just the prior. **E. Simulations with Metacognitive Noise for Explicit Confidence Only.** (Top) Decision level results given optimal prior weighting but with metacognitive noise added only for explicit confidence, which does not impact the decision level. (Bottom) Confidence level results given equal

underweighting of the prior between decisions and confidence, and added metacognitive noise just in explicit confidence. For all simulations shown here, substantial metacognitive noise was added, approximately equivalent to an MRatio of 0.5.

In sum, we show that metacognitive noise cannot provide an alternative explanation for the results, as none of the several implementations of metacognitive noise that we tried can capture the difference between conditions in target decision accuracy, nor can they capture the asymmetry between the decision and confidence levels. We also note that Exp. 3, described below, now additionally demonstrates that our findings hold even when using a prior that is not based on a confidence computation, and hence not susceptible to metacognitive noise.

2.3. - Is it possible that the apparent discrepancy, in that confidence underweights the prior LESS than decisions do, could be due to metacognitive inefficiencies? That is, is it the case that confidence optimally uses the (inefficiently estimated) prior at the noise ceiling due to confidence leak from lead stimulus to target stimulus? I am not sure I am being clear here (and not sure exactly how this would work), but it seems there might be some effect of ‘confidence leak’ between lead and target stimulus that could interact with the measurement of how effectively the prior is used. There also feels like a conceptual differentiation here, which is that the ‘kind’ of information resulting from estimating the prior from the lead stimulus (i.e., decision confidence) is the same ‘kind’ of information used in the confidence judgment in the target stimulus. Thus, perhaps it’s not so much that the prior is underweighted, but that the use of the prior is corrupted when translating between type 2 and type 1 reports. Much like we often assume that type 2 noise corrupts signals when they flow from the type 1 to the type 2 space, here the authors are asking whether information ‘flowing’ from the type 2 to the type 1 space can be used optimally; if the same kind of ‘translation’ noise is corrupting the information regardless of whether it goes from type 1 → type 2 or type 2 → type 1, then there might again be some relationship between underweighting of prior information in the decision and metacognitive efficiency (M-ratio), as discussed in the previous bullet. Perhaps the authors could consider and discuss these issues.

We consider two key points mentioned here: (1) how confidence leak from the lead confidence to the target confidence might interact with our findings, and (2) whether translation noise when moving the prior between the type 1 and type 2 space can account for our results. Both this confidence leak model and the possibility of translation noise are now discussed in the manuscript, in a new section on alternative explanations of the results (pg. 20-21), and in depth in Supplementary Materials.

(1) A form of confidence leak from the internal lead confidence to the explicit target confidence is possible, and interesting to consider. We now include the following analysis of this in Supplementary Materials:

“The use of a prior that is itself an internal confidence value leads to the possibility that there is an impact of the internal confidence on the explicit confidence, beyond the prior strength. This would be in line with the finding that confidence can “leak” from one trial to the next⁶⁸. Here, it is possible that internal confidence from the lead decision likewise biases the explicit confidence. This would not, however, impact the weighting of the prior in the target decision, so it could not explain the underweighted prior observed at the decision level. In confidence, this confidence leak model predicts that the higher internal lead decision confidence in the Stronger-Prior condition would leak to produce more extreme target decision confidence reports. However, given the underweighted prior in decisions, simulations from a confidence leak model showed that the effects on confidence cannot explain the asymmetry between decisions and confidence that we find, either at moderate values based on previous findings⁶⁸ (Fig. S10B-C), nor at extreme values (Fig. S10D). This also cannot capture the pattern of results when it interacts with metacognitive noise (Fig. S10C-D). Confidence leak was simulated by setting the final explicit confidence rating to be a weighted average between the target confidence (before any leak) and the internal lead confidence, and strength of the leak was manipulated by adjusting the relative weight of lead confidence in this average.”

Figure S10. Confidence Leak Simulations. Simulations of confidence patterns given equal underweighting of priors in decisions and confidence (w_{choice} and $w_{conf} = 2.17$), with an additional confidence leak effect. **A. Data.** The true confidence patterns for correct versus incorrect trials per

condition and posterior information level. **B. Moderate Confidence Leak.** Simulations of confidence without metacognitive noise, and with a moderate confidence leak strength such that one unit of confidence increase on the lead decision predicts an increase of target confidence of approximately 0.25. **C. Moderate Confidence Leak with Metacognitive Noise.** Simulations of confidence with the same amount of confidence leak as in (B), but with added metacognitive noise. **D. Extreme Confidence Leak.** Simulations demonstrating an extreme confidence leak, to show the direction in which this shifts results. This also includes the same metacognitive noise as in (C). Together, this reveals that confidence leak cannot account for the pattern of results – even in interaction with metacognitive noise or at extreme values – which can instead be captured by asymmetrical weighting of prior information.

(2) *It could be that the prior gets corrupted by noise not only when moving from the type 1 to the type 2 space, as in the case of metacognitive noise (see response to 2.2 above), but also when being translated back to the type 1 space for use in the target decision. This is discussed in Supplementary Materials, copied below:*

“Because of the nature of the prior as an internal confidence value in the dual-decision task, it is possible that its computation exists in a Type 2, metacognitive processing space. If this is the case, it may also be that in order to make use of that prior in the target decision, it must be translated back to the Type 1 decision space. This translation could itself also be corrupted by a “translation” noise, similar to the metacognitive noise that occurs when moving from the Type 1 to the Type 2 space. Translation noise will have the same impact as metacognitive noise, making the prior estimation noisier and less reflective of the true prior, but not consistently too weak or too strong. Simulations (not included here) have revealed that, as with adding metacognitive noise, (a) target decision accuracy decreases and (b) confidence patterns get less informative following correct vs incorrect target decisions, but critically it does not interact with our conditions. We conclude that it cannot provide an alternative explanation for our results.”

We note that the results of these simulations are not included because they are essentially identical to the results of the metacognitive noise analysis.

In sum, while interesting to consider, neither confidence leak or translation noise can account for the pattern of results. We thank the reviewer for raising these considerations, which enrich our model interpretations.

*Rahnev, D., Koizumi, A., McCurdy, L. Y., D’Esposito, M., & Lau, H. (2015). Confidence leak in perceptual decision making. *Psychological science*, 26(11), 1664-1680.*

2.4. - There is a lot of work on the “right” kind of metacognitive noise/corruption: normally-distributed type 2 noise, multiplicative internal noise, log-normally distributed internal noise, use of decision-congruent evidence while downweighting/ignoring decision-incongruent evidence... basically, as the authors I know are aware, this field is a bit of a mess right now. Nevertheless, it feels important to discuss the potential implications of these other kinds of noise/metacognitive inefficiency/estimation

assumptions on the results. The model fitted has an internal noise factor fitted to each subject (lines 305-306, for example)

The noise assumption made here in the main modeling analyses, and in fitting the internal noise factor, was that first-order internal processing noise is Gaussian, or in other words that the internal signals will be normally distributed around the external signal strength. This assumption was made in order to stay in line with the basic assumptions of SDT. We did not make strong assumptions about metacognitive noise in our modeling. However, we agree that this must also be considered for a complete picture of decision confidence and metacognitive processing. We have now considered in more depth the different ways in which metacognitive noise might factor into our model and results, to work towards this more complete picture. In addition, following the reviewer's suggestion, here we consider the impact of metacognitive noise that is not necessarily normally distributed. We simulated log-normally distributed metacognitive noise following the same possible implementations as above (see Response 2.2). Like normally-distributed metacognitive noise, we found this not to impact our conditions in any of those implementations. Hence, while our work cannot clarify what the "right" kind of metacognitive noise is, we feel that we can contribute to another aspect of the picture, namely better understanding precisely how different kinds of information – such as information from priors, which are thought to be quite ubiquitous to our processing – impact confidence computations.

Reviewer #3 (Remarks to the Author):

In this manuscript, Constant and colleagues report on empirical and modelling investigations of decisions and metacognitive judgments. Using an elegant dual-decision task, adapted from previous work, they independently manipulate the strength of priors and likelihoods on which decisions and confidence judgments are based. In two experiments (N = 21 + 25), they report that priors are underweighted in decisions, but used to a greater extent in confidence judgments about those decisions. They finally proceed to account for this result with a Bayesian model.

There is a lot to like in the manuscript: the task is very elegant, it is combined with straightforward hypotheses, that are tested thanks to a clear and sharp operationalization (experimental manipulation). The results seem robust, as they replicate in the 2 (close to identical) instantiations of the task.

My potential enthusiasm is nonetheless tempered by several conceptual and methodological concerns which should be addressed before I can recommend the manuscript for publications.

Main concerns

3.1. First, the manuscript is written in a very sharp/straightforward way. This makes it a pleasure to read but comes at the cost of oversimplifications and overgeneralization. In particular, although I find the task very elegant, I cannot help but worry that some (if not most) of the main conclusions/findings are very specific/idiosyncratic of the present experimental setup (dual-task), where the prior is defined by the performance of a first decision task. It seems to me that if one just wanted to test the weight of priors vs likelihood weights in decision and confidence judgments, another (more?) straightforward way would be a single-task that includes an exogenous cue, which provides an exogenous (rather than endogenous) prior evidence (as per instruction, e.g. a dot whose color correlates with probability of the true state of the world being A/B). This brings the question: what is specific vs generalizable in the current findings, in this dual task? Of course, it is not my role to impose a new experiment (though it would be very informative). But I think the authors should, at the very least, 1) better delineate the specificity of the current setup (dual task) and how this can help the reader better interpret/conceptualize the results, and 2) be more explicit about the (potential) specificity of their results to the current setup.

We thank Reviewer 3 for their positive comments about the elegance of the task, and address their concerns about its possible idiosyncrasies below.

We acknowledge that the dual-decision task is not the most common task structure in work manipulating priors. We chose this structure for a few key reasons. First, it allowed us to create the matched conditions, which, critically, was necessary in order to give us a way to quantitatively assess the behavioural results without relying on the model. Second, having the prior and likelihood both come from a dot motion stimulus made it so that they were the same 'kind' of information, allowing us to make the same assumptions about both – for example in terms of how people interpret them.

However, to address the reviewer's concern about the generalizability of our results, we decided to run a third experiment, using exogenous probabilistic cues as priors. Namely, on each trial participants were given an explicit cue informing them of the probability that the coherent motion of the stimulus would be going either right or left (counterbalanced across participants). These cues corresponded truthfully to either a probability of 0.6, 0.7, 0.8, or 0.9. They then viewed the stimulus and made their decision, and then rated confidence on a sliding scale from 50-100%. They were instructed to use these cues to help them, and were given as long as they wanted to read and internalize the cue, as well as to make their decision and confidence rating. There were additionally some blocks of trials without cues, which corresponded to a probability of 0.5 of rightward/leftward stimuli, or a flat prior. These trials were used to fit decision bias, internal noise, and metacognitive noise (see R1#1.1 and R2#2.2), which was necessary for the full modeling of the trials under the influence of an informative prior. The model from Exp. 1 and 2 was adapted to consider the probabilistic priors in Exp. 3, and the same parameters were fit, including the weighting parameter of the prior in decisions

(w_{choice}), the weighting parameter of the prior in confidence (w_{conf}) and the confidence bias parameter (b).

Crucially, we found the results to generalize to this task: The prior was underweighted in decisions and then used to a greater extent in confidence (see the new Figure 7, copied from the manuscript below).

Experiment 3 Results. A. Decision Level Predictions. The probability of choosing right given each level of rightward prior was simulated with three levels of prior weighting: optimal weighting ($w_{choice} = 1$), overweighting ($w_{choice} = 0.5$), and underweighting ($w_{choice} = 2$). The model predicts that weaker weighting of prior information in the decision leads to a weaker relationship between the strength of the rightward prior and the probability of choosing right. **B. Decision Level Results.** The simulations from the fit model are shown against the data, along with the optimal weighting as a reference. This reveals a weaker slope of the probability of rightward decisions across rightward prior strengths compared to optimal, suggesting an underweighting of priors in decisions. **C. Confidence Level Predictions.** Confidence given each level of rightward prior was simulated with three levels of prior weighting relative to underweighting at the decision level ($w_{choice} = 2$): equal weighting ($w_{conf} = 2$), stronger weighting ($w_{conf} = 1$), and weaker weighting ($w_{conf} = 4$). The model predicts that weighting of the prior in confidence influences the curvature of the relationship between confidence and the strength of the prior. To highlight that curvature, confidence is shown normalized to the flat prior condition (Prior Strength = 0.5) as a baseline by taking the difference in mean confidence to that baseline. **D. Confidence Level Results.** The simulations from the fit model are shown against the data. Additionally, to demonstrate asymmetrical prior weighting with the decision level, symmetrical weighting ($w_{conf} = w_{choice}$) is shown as a reference. This reveals stronger concavity across rightward prior strengths compared to equal weighting, indicating stronger weighting of the prior in confidence than in decisions. **E. Posterior distribution for w_{choice} .** The posterior distribution for the group mean parameter of the weighting of prior information in the decision, w_{choice} . The blue shaded region shows the 89% credible interval and the vertical black dashed line reflects optimal weighting of the prior in the decision ($w_{choice} = 1$). A weight above 1 captures overestimation of the variance and hence underweighting of the prior. **F. Posterior distribution for w_{conf} and b .** The top posterior distribution is for the group mean parameter of w_{conf} . The lower posterior distribution is for the group mean parameter of b . The blue shaded regions show the 89% credible intervals and the vertical black dashed line corresponds to the parameter values of an optimal observer. **G. Posterior Group Difference Distribution of $w_{choice} - w_{conf}$.** The posterior distribution for the difference in the group mean parameters w_{choice} and w_{conf} . The blue shaded region shows the 89% credible interval

and the vertical black dashed line reflects no difference in the two parameters ($w_{choice} - w_{conf} = 0$). 0 is just excluded from the 89% credible interval, suggesting w_{choice} and w_{conf} to be credibly different from one another.

This demonstrates that the patterns found here are not unique to the dual-decision setup. We argue that this makes our conclusions highly relevant to the tasks used throughout the literature, and to interpreting other results. For example, extensive other work has found an underuse of priors in decisions, often reported as an overly conservative criterion bias (Ackermann & Landy, 2015; Bang & Rahnev, 2017; Botzer et al., 2010, 2013; Morales et al., 2015; Murrell, 1977; Phillips & Edwards, 1966; Rahnev & Denison, 2018; Ulehla, 1966). Here, we replicate this, but critically, we show that the prior information that is ignored in decisions is still processed and used to a greater extent at the metacognitive level. This gives an important new layer of insight to the existing literature, suggesting that the information that might appear to be lost is still processed, monitored, and can still impact behaviour, but that this occurs at a different level than perceptual decisions.

The generalization of our conclusions to this new task structure also importantly demonstrates that the pattern of results does not emerge due to other features of the dual-decision setup, such as the nature of the prior as an internal confidence computation. We have added a description of the results and methods of Experiment 3 on pages 16-18 and 26 respectively.

- Ackermann, J. F., & Landy, M. S. (2015). Suboptimal decision criteria are predicted by subjectively weighted probabilities and rewards. *Attention, Perception, & Psychophysics*, 77(2), 638–658. <https://doi.org/10.3758/s13414-014-0779-z>
- Bang, J. W., & Rahnev, D. (2017). Stimulus expectation alters decision criterion but not sensory signal in perceptual decision making. *Scientific Reports*, 7(1), Article 1. <https://doi.org/10.1038/s41598-017-16885-2>
- Botzer, A., Meyer, J., Bak, P., & Parmet, Y. (2010). User settings of cue thresholds for binary categorization decisions. *Journal of Experimental Psychology: Applied*, 16, 1–15. <https://doi.org/10.1037/a0018758>
- Botzer, A., Meyer, J., & Parmet, Y. (2013). Mental Effort in Binary Categorization Aided by Binary Cues. *Journal of Experimental Psychology: Applied*, 19, 39–54. <https://doi.org/10.1037/a0031625>
- Morales, J., Solovey, G., Maniscalco, B., Rahnev, D., de Lange, F. P., & Lau, H. (2015). Low attention impairs optimal incorporation of prior knowledge in perceptual decisions. *Attention, Perception, & Psychophysics*, 77(6), 2021–2036. <https://doi.org/10.3758/s13414-015-0897-2>
- Murrell, G. A. (1977). Combination of evidence in a probabilistic visual search and detection task. *Organizational Behavior & Human Performance*, 18, 3–18. [https://doi.org/10.1016/0030-5073\(77\)90015-0](https://doi.org/10.1016/0030-5073(77)90015-0)
- Phillips, L. D., & Edwards, W. (1966). Conservatism in a simple probability inference task. *Journal of Experimental Psychology*, 72, 346–354. <https://doi.org/10.1037/h0023653>
- Rahnev, D., & Denison, R. N. (2018). Suboptimality in perceptual decision making. *The*

3.2. Confidence judgments seem, overall, very in-sensitive to the overall amount of (posterior) evidence (Figure 2C, 6B). Would a model without main effect of condition (stronger Lead/Target) still replicate the commonly found effect of choice accuracy and interaction between accuracy and evidence on confidence (so called X-pattern)? Also, I'm wondering to what extent it would make sense to reproduce the panel of Figure 2B and 2C (accuracy and confidence) as a function of prior strength and likelihood strength (rather than just posterior strength), to provide new dimensions to evaluate pattern of results to validate/falsify the tested models.

We have now explored further the relationship between confidence and posterior information. When collapsing across conditions – as suggested by the reviewer – we do actually find that we replicate the folded-X pattern (see figure below). This analysis is now included in the manuscript (pg. 7):

“We built a linear mixed effects model on confidence with fixed effects of posterior level (L, M, H), target response accuracy (Correct, Incorrect), and their interaction. This revealed the expected interaction between posterior level and response accuracy, $F(2,14842)=24.5$, $p<0.001$, $BF_{10}=2.84 \times 10^6$, replicating the folded-X confidence pattern.”

We have now included below the results in Fig. 2B and 2C but shown as a function of lead and target level (with the resulting posterior level still shown through color-coding). Although the original panels were organized by posterior level and condition, this still showed all of the same combinations of lead and target level – the low posterior level contains $lead_{low} + target_{medium}$ (Stronger-Target condition) as well as $lead_{medium} + target_{low}$ (Stronger-Lead condition); the medium posterior level contains $lead_{low} + target_{high}$ (Stronger-Target condition) as well as $lead_{high} + target_{low}$ (Stronger-Lead condition); and the high posterior level contains $lead_{medium} + target_{high}$ (Stronger-Target condition) as well

as $lead_{high} + target_{medium}$ (Stronger-Lead condition). Hence, the same information is captured in the original panels, and we wanted to organize it that way to highlight the differences between conditions.

3.3. Although a normative model is always an excellent benchmark, I am actually a bit frustrated that the authors did not explore other ways to explain their findings.

We started with this normative model, exactly as the reviewer points out, as a benchmark with which to quantify and understand how different types of information are weighted in decisions and confidence. We agree though, that it is important to consider in more depth what other models or heuristics can explain the asymmetry and suboptimality that we find from the normative model benchmark. In response to this comment, and in line with the suggestions from Reviewers 1 and 2, we have now added a section to the manuscript considering our findings in the context of several alternative models. These include the confirmatory choice model that the reviewer highlights below, different metacognitive noise models (including normally-distributed and log-normally distributed metacognitive noise), and a confidence leak model. We additionally discuss these alternative explanations in depth in Supplementary Materials. In brief, we found that none of these alternative models could explain our pattern of results, confirming that the best interpretation is that priors are underweighted in discrimination decisions but used to a greater extent in confidence.

For instance, it could very well be that there are non-additive interactions between the two-steps of dual task. Notably, decision-makers are known to behave in a confirmatory/motivated way: basically, they process evidence, make decisions and evaluate confidence in a way that is confirmatory see e.g. (Rollwage et al., 2020; Talluri et al., 2018, 2021) – this actually also connects/extends to the topic of choice-congruent evidence mentioned by the authors in the introduction. I'm therefore wondering whether such a process could happen here, where the target-likelihood could be evaluated in an asymmetric way, to confirm the lead-choice (creating an asymmetry between confidence in correct versus incorrect, that also depends on the condition lead/target first).

We thank the reviewer for highlighting the interesting consideration that participants may act in a confirmatory way in the dual-decision task, which could interact in subtle ways with our findings. Simulations revealed that this could produce patterns in which there is higher accuracy in the Stronger-Target condition. However, it naturally also predicts that participants are more likely to make target decisions in line with the lead decision. This is not what we find in our data, and in fact we see a small tendency towards the opposite pattern, which we refer to below as an opposing choice bias, with participants more often choosing “right” on target decisions following leftward lead choices. We now include in the Supplementary Materials the following, confirming that neither a confirmatory nor an opposing choice bias provides an alternative explanation of our results:

“We find that participants do not combine both sources of evidence (prior and likelihood) optimally when forming their target decision. This is in line with previous work that has found non-additive or biased combinations of evidence. For example, several studies have found a confirmation bias in the way that evidence accumulates over time, with later evidence being biased towards supporting the decision already made, even if that decision was internal^{43,64,65}. If that occurs in our paradigm, we would expect the second decision to be biased towards confirming the first decision. Although this is not in line with the rule we gave participants, it is possible that they gathered evidence in this way. To assess this, we investigated whether participants performed better when the direction of the target stimulus matched the lead, which would be expected if they are biased to confirm their lead decisions. This revealed participants not to have such a confirmatory bias, and in fact to have a small bias in the opposite direction, or a slight repulsion effect, with participants being more likely to choose right on targets following a leftward lead decision. So, this confirmatory evidence accumulation does not explain our pattern of results. Additionally, at the confidence level this confirmatory behaviour would lead to higher confidence following confirmatory decisions, compared to opposing decisions (Figure S8B), which we do not find in the data (Figure S8A). Importantly, though we find a repulsion effect at the decision level, the confidence patterns also do not fit what would be expected of an opposing choice bias, which produces lower confidence following confirmatory decisions, compared to opposing decisions (Figure S8C). Hence, this opposing choice effect also cannot capture our pattern of results.”

Figure S8. Predicted Confidence with Confirmatory and Opposing Choice Biases. A. Data. The true mean confidence following correct and incorrect trials per posterior information level, split by whether the target decision was the same as or was different from the lead decision. Error bars indicate SD across all participants. **B. Simulated Confidence with Confirmatory Choice Bias.** Simulated confidence with a confirmatory bias such that target evidence is biased to be more strongly in line with the lead decision. This leads to higher confidence (following both correct and incorrect trials) when the target decision is the same as (confirms) the lead decision. **C. Simulated Confidence with Opposing Choice Bias.** Simulated confidence with an opposing choice bias such that target evidence is biased away from the direction of the lead decision. This leads to higher confidence (following both correct and incorrect trials) when the target decision goes against (opposes) the lead decision.

Together, the results cannot be explained by either a confirmatory choice bias or an opposing choice bias. Additionally, the results from the new Experiment 3 demonstrate that the results hold even outside of a dual-decision setup, suggesting that they are not simply due to non-additive interactions between the two decisions.

This may be a direction to explore, notably to explain one of the current model main misfit, which is a systematic underestimation of confidence judgments (Figures 4B, 6B)

As we now better explain on pg. 10-11, the model's systematic underestimation of confidence judgments was predictable due to the particular model simplification we made. Without the simplification, confidence is not systematically underestimated (see Response 1.2). However, this simplification was necessary in order to fit the model hierarchically, and despite the predictable underestimation of confidence, we ensured that this could not interfere with our interpretation of the weighting parameters. For a more in depth discussion of this point, see Response 1.2, above.

Other

3.4. Regarding model recovery (Figure S4), I encourage the authors to commit to a more classic approach: define a model-comparison winning criteria, and produce proper confusion matrices.

We have now implemented this approach and show confusion matrices (now Figure S5 due to other added figures).

3.5. This might seem completely trivial, but the fact that step-1 correct choices are deterministically associated with the right direction as the step-2 creates could be somewhat “dangerous”, if there is any decision bias toward one side. Why not balancing this (even if just between-subject)?

It is true that it is important to consider this imbalance, since it can interact with decision bias. However, we did account for it in our analyses. We took the akaike-weighted combination of four psychometric functions fit to the control trials of each individual participant in order to measure their decision bias, and then incorporated that bias into all further modeling analyses. We also included that fit bias level as a regressor in the linear mixed effects models, which did not change any of our conclusions.

We chose to account for decision bias this way rather than including both directions of the dual-decision rule because we predicted that, due to the likely presence of decision bias, we would have to analyze the directions separately. This could potentially decrease the trial counts available for fitting the model, which we wanted to avoid.

However, we counterbalanced this between subjects in Experiment 3, as the reviewer points out, and we thank the reviewer for the suggestion.

Rollwage, M., Loosen, A., Hauser, T. U., Moran, R., Dolan, R. J., & Fleming, S. M. (2020). Confidence drives a neural confirmation bias. *Nature Communications*, 11(1), Article 1. <https://doi.org/10.1038/s41467-020-16278-6>

Talluri, B. C., Urai, A. E., Bronfman, Z. Z., Brezis, N., Tsetsos, K., Usher, M., & Donner, T. H. (2021). Choices change the temporal weighting of decision evidence. *Journal of Neurophysiology*, 125(4), 1468–1481. <https://doi.org/10.1152/jn.00462.2020>

Talluri, B. C., Urai, A. E., Tsetsos, K., Usher, M., & Donner, T. H. (2018). Confirmation Bias through Selective Overweighting of Choice-Consistent Evidence. *Current Biology*, 28(19), 3128-3135.e8. <https://doi.org/10.1016/j.cub.2018.07.052>

REVIEWERS' COMMENTS

Reviewer #1 (Remarks to the Author):

To address why the model fits for confidence are off, the authors demonstrated using recovery analyses that, if anything, this makes their analyses more conservative. I found the reasoning a bit odd (if the model doesn't fit the data well, I think that could be an issue regardless of the direction of the misfit), but the inclusion of the full (non-hierarchical) model was satisfying and convincing. It was not very clear to me why the authors decided to keep the restricted hierarchical in the main text instead of the full model.

The authors now include the full results of Experiment 2, which I think is good. Although the results are not as perfect as one could hope for (which is fine, data are noisy), the inclusion of a new Experiment (3) tipped the balance for me.

Good job on a very good interesting paper.

Reviewer #2 (Remarks to the Author):

The authors have done an excellent job addressing my concerns and I am satisfied with the edits. They also ran a whole third experiment, which strengthened the findings. I want to thank the authors for so thoroughly exploring the comments I raised. I think this is a timely and important contribution to the field and look forward to seeing it published.

Reviewer #3 (Remarks to the Author):

I think the author did a very serious job in their attempt to address most comments – including adding a new experiment to address one of my own concern. There might still be some non-ideal patterns of results (e.g. confidence in Exp. 2) or possible alternative explanations, but I believe that the reported findings are now established robustly-enough to enable the publication of this very elegant and thought-provoking manuscript.

REVIEWER COMMENTS

Reviewer #1 (Remarks to the Author):

To address why the model fits for confidence are off, the authors demonstrated using recovery analyses that, if anything, this makes their analyses more conservative. I found the reasoning a bit odd (if the model doesn't fit the data well, I think that could be an issue regardless of the direction of the misfit), but the inclusion of the full (non-hierarchical) model was satisfying and convincing. It was not very clear to me why the authors decided to keep the restricted hierarchical in the main text instead of the full model.

We appreciate the comment that the inclusion of the full, non-hierarchical model makes the interpretations more convincing, so we have now included those results in the main text (line 272-281; 323-333). Figure 3B and 4B now show the model results from the full model, no longer underestimating confidence in 4B. We have moved those model results from the simplified model into the Supplementary Information.

The authors now include the full results of Experiment 2, which I think is good. Although the results are not as perfect as one could hope for (which is fine, data are noisy), the inclusion of a new Experiment (3) tipped the balance for me.

Good job on a very good interesting paper.

Reviewer #2 (Remarks to the Author):

The authors have done an excellent job addressing my concerns and I am satisfied with the edits. They also ran a whole third experiment, which strengthened the findings. I want to thank the authors for so thoroughly exploring the comments I raised. I think this is a timely and important contribution to the field and look forward to seeing it published.

Reviewer #3 (Remarks to the Author):

I think the author did a very serious job in their attempt to address most comments – including adding a new experiment to address one of my own concern. There might still be some non-ideal patterns of results (e.g. confidence in Exp. 2) or possible alternative explanations, but I believe that the reported findings are now established robustly-enough to enable the publication of this very elegant and thought-provoking manuscript.

We agree with all reviewers that Exp. 3 has strengthened the robustness of the findings, and we thank all reviewers again for the helpful, insightful input.